# CoPatch: Zero-Shot Referring Image Segmentation by Leveraging Untapped Spatial Knowledge in CLIP

## Abstract

Spatial grounding is crucial for referring image segmentation (RIS), where the goal of the task is to localize an object described by language. Current foundational vision-language models (VLMs), such as CLIP, excel at aligning images and text but struggle with understanding spatial relationships. Within the language stream, most existing methods often focus on the primary noun phrase when extracting local text features, undermining contextual tokens. Within the vision stream, CLIP generates similar features for images with different spatial layouts, resulting in limited sensitivity to spatial structure. To address these limitations, we propose CoPatch, a zero-shot RIS framework that leverages internal model components to enhance spatial representations in both text and image modalities. For language, CoPatch constructs hybrid text features by incorporating context tokens carrying spatial cues. For vision, it extracts patch-level image features using our novel path discovered from intermediate layers, where spatial structure is better preserved. These enhanced features are fused into a clustered image–text similarity map, `CoMap`, enabling precise mask selection. As a result, CoPatch significantly improves spatial grounding in zero-shot RIS across Ref-COCO, RefCOCO+, RefCOCOg, and PhraseCut (+ 2–7 mIoU) without requiring any additional training. Our findings underscore the importance of recovering and leveraging the untapped spatial knowledge inherently embedded in VLMs, thereby paving the way for opportunities in zero-shot RIS.

## 1 Introduction

Understanding spatial concepts is a fundamental cognitive ability that enables humans to interact with complex environments for real-life tasks (Wolbers & Hegarty, 2010). While humans naturally acquire spatial grounding ability (Wolbers & Hegarty, 2010; Piaget, 1952; Newcombe & Huttenlocher, 2000), current vision-language models (VLMs) still face significant challenges in this area. In particular, CLIP (Radford et al., 2021), one of the most widely used pretrained foundational VLMs, shows limited ability to perform even basic spatial or positional understanding (Tong et al., 2024; Tang et al., 2023; Peng et al., 2024; Kamath et al., 2023; Du et al., 2024; Chatterjee et al., 2024; Huang et al., 2023). This limitation is especially critical in real-world applications that rely heavily on fine-grained spatial grounding, such as robotic object manipulation (Billard & Kragic, 2019; Jang et al., 2006) and autonomous driving (Chen et al., 2024; Kim et al., 2024).

A particularly demanding testbed for spatial grounding is *zero-shot* referring image segmentation (RIS). In this task, the model needs to accurately segment the object *referred* by a natural language expression, without any task-specific training, utilizing off-the-shelf pretrained segmentation models. This requires a pretrained CLIP to comprehend both the semantic meaning and the spatial location of the referred object (Subramanian et al., 2022; Yu et al., 2023; Suo et al., 2023; Wang et al., 2025; Liu & Li, 2025; Ni et al., 2023; Han et al., 2024; Lüddecke & Ecker, 2022). The challenge becomes even more significant when multiple similar objects appear in different locations (Nagaraja et al., 2016; Mao et al., 2016; Wu et al., 2020) (Figure 1). Although the CLIP-based methods have made significant progress in aligning visual and textual semantics (An et al., 2025a), they continue to struggle with precise spatial grounding.

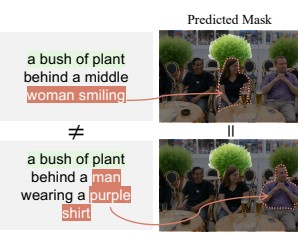 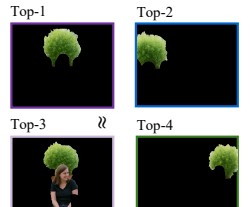 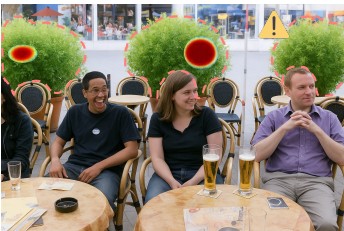

(a) Context Token Disregard     (b) Overlapping Masks     (c) Incomplete Localization

Figure 1: **Limitations of existing zero-shot RIS approaches**. Prior works lack in (a) incorporating contexts containing important cues for local-level text feature extraction, (b) filtering out top masks that may contain overlapping contents, and (c) spatially grounding all the target objects.

More specifically, existing zero-shot RIS approaches fail to encode spatial information via original feature extraction pathways for both texts and images (Chng et al., 2024). On the text side, their focus is on the primary noun phrase when extracting local-level text features. However, this could leave out important cues, namely, *context tokens*, that may appear after the primary noun chunk. This often leads to identical mask outputs for semantically different expressions, as shown in Figure 1a. On the image side, previous methods directly use the original CLIP image features, discarding *spatial layout* information maintained until the final layer. As a result, masked images with similar, overlapping semantics are selected in the top candidate masks (Figure 1b), reducing the reliability of the final mask selection process. These limitations underscore the need to leverage untapped model components to enhance zero-shot RIS ability for CLIP.

In this work, we present COPATCH, a training-free framework that unlocks the underexplored spatial understanding ability of CLIP for RIS. Our approach has two main components: First, we construct hybrid text features that incorporate *context tokens* carrying useful spatial cues. This textual guidance enables the model to capture subtle spatial distinctions of the same object names in referring expressions. Second, we devise an expressive *patch-level spatial map* through a newly identified pathway in CLIP's visual encoder. This is motivated by our observation that the intermediate layers of the visual encoder can better preserve spatial information than the last layer. By combining these two sources, we generate a context-aware spatial map, CoMap, which reranks top candidate masks with greater spatial precision.

The novelty of COPATCH lies in its ability to enhance both text and image modalities without any additional training. Unlike gradient-based (Selvaraju et al., 2016) (Figure 1c) or attention-based (Li et al., 2023; Bousselham et al., 2024) localization methods, our approach is computationally efficient while offering stronger zero-shot spatial grounding. This shows that precise zero-shot segmentation can be achieved not by redesigning models or costly fine-tuning, but by strategically leveraging the underused capacities already present in pretrained foundational models.

To summarize, our contributions are as follows: 1) **Novel pathway for informative spatial map generation**. We discover a new computation-efficient pathway in CLIP's visual encoder. This enables the extraction of a context-aware spatial map, CoMap, with a single forward pass without recomputations. 2) **Effective mask scoring strategy**. We propose a novel scoring framework that leverages CoMap to effectively rerank top candidate masks, ensuring that the most semantically aligned, non-overlapping masks are retained. This significantly improves the reliability of the final mask selection process. 3) **SoTA zero-shot RIS performances**. Our proposed COPATCH achieves mIoU scores of 55.62, 55.38, and 54.08, in the RefCOCO (Nagaraja et al., 2016), RefCOCO+ (Nagaraja et al., 2016), and RefCOCOg (Mao et al., 2016) test sets, outperforming the current SoTA (Liu & Li, 2025) by +4.03p, +2.01p, and +4.95p. We hope our work paves the future work in proposing various zero-shot RIS methods by extracting enhanced inherent spatial knowledge from VLMs.

## 2 RELATED WORKS

**Referring Image Segmentation.** Referring Image Segmentation (RIS) aims to segment image regions described by text expressions. Early supervised methods rely on mask annotations to learn dense cross-modal correspondences. LAVT (Yang et al., 2022) introduces a language-aware vi-

sion transformer that employs early fusion and stacked cross-modal attention for feature refinement. CRIS (Wang et al., 2022b) adds a CLIP-driven decoder trained with a contrastive loss to enhance text–pixel alignment. RISCLIP (Kim et al., 2023b) adapts CLIP features through parameter-efficient fusion layers. ETRIS (Xu et al., 2023) inserts lightweight adapter modules between frozen CLIP encoders, achieving competitive accuracy with few additional parameters. Prompt-Driven RIS (Shang et al., 2024) injects instance-aware prompts into CLIP and refines masks using SAM (Kirillov et al., 2023). UniNeXt (Lin et al., 2023) aims for unification, integrating RIS with other vision tasks using shared transformer heads. MagNet (Chng et al., 2024) aligns the image and text encoders, especially at a fine-grained pixel and word levels. Ito (2025) introduces a *trained* instance segmentation module to better bridge the CLIP and SAM models.

To mitigate the high cost of pixel-level annotations, weakly-supervised methods emerged, relying only on image-text pairs. Techniques include enforcing text-region consistency (Strudel et al., 2022). Moreover, the prior approaches maintain intra-chunk semantic coherence (Lee et al., 2023), connect regions to linguistic cues via shatter-and-gather mechanisms (Kim et al., 2023a), and mine text-region interactions (Liu et al., 2023). Pseudo-supervised approaches, such as Pseudo-RIS (Yu et al., 2024), further bridge the gap by leveraging foundation models to generate high-quality pseudo-masks for training, eliminating the need for manually generating ground-truth masks.

**Zero-shot Referring Image Segmentation.** Recent zero-shot RIS methods leverage pretrained vision-language models (VLMs) to avoid costly pixel-wise annotations and training with the following standard pipeline: 1) generating candidate masks using a pretrained segmentation model (Kirillov et al., 2023; Cheng et al., 2022; Liang et al., 2023; Wang et al., 2022a), 2) computing a similarity score between each masked image and the referring expression, and 3) selecting the mask with the highest score as output. To improve zero-shot mask scoring (the second stage), various methods have been proposed: ReCLIP (Subramanian et al., 2022) uses a heuristic spatial relation resolver to score the isolated mask proposals by calculating the probability for each noun chunk node. CLIPSeg (Lüddecke & Ecker, 2022) segments the image using a set of prompts at inference time. Several fully training-free comparison baseline methods include: a) **Region Token** (Li et al., 2022a) method calculates the cosine similarity between spatial feature per pixel and text embeddings for extracting a final mask (Suo et al., 2023). b) **CLIPSurgery** (Li et al., 2023) and c) **Cropping** use the average activation score (Suo et al., 2023) and cosine similarity between cropped image features and text features (Yu et al., 2023) for scoring masks. d) **Global-Local** uses the cosine similarity between global-local image features and text features (Yu et al., 2023). e) **TAS** augments the mask scores using features of negative texts generated by an external captioner, BLIP-2. f) **HybridGL** (Liu & Li, 2025) proposes a new image/text feature extraction strategy by fusing global image features into local image features (G2L) and using the cosine similarity between the hybrid G2L image and global-local text features.

Our method aligns more similarly to several recent zero-shot RIS methods that use a spatial map as a guide for the mask scoring step. For instance, **Ref-Diff** (Ni et al., 2023) uses a diffusion model to generate their own spatial map and uses a weighted sum of images multiplied by the positional bias matrix and mask proposals. **CaR** (Sun et al., 2024) uses Grad-CAM as the spatial map to generate mask proposals and filters the masks using similarity between each query (that may contain irrelevant texts) and masked images. **IteRPrimE** (Wang et al., 2025) uses an interpolated Grad-CAM heatmap generated using the primary word to guide the model for the candidate mask selection. **HybridGL** (Liu & Li, 2025) uses pretrained GEM (Bousselham et al., 2024) as the spatial map and reranks the initial mask scores with their proposed spatial *relation* and *coherence* guidance. Our CoPatch is distinguished from HybridGL because it uses different spatial *relation* guidance using our proposed spatial map, but incorporates their spatial *coherence* method to boost the final performance (*see* Method and Ablation Study in Sections 3.2 and 4.3).

## 3 CoPatch: Context Token-Patch Level Embedding Synergy

Our proposed framework, CoPatch, is designed to recover CLIP's latent spatial grounding ability by jointly enhancing text and image representations (Figure 2). This framework has two essential components: 1) Hybrid text features with *context tokens*, which preserve spatial cues in referring expressions (Section 3.1), and 2) context-aware spatial map, CoMap, generated by our enhanced text and spatial-aware image features (Section 3.2). Unlike prior zero-shot RIS methods that rely

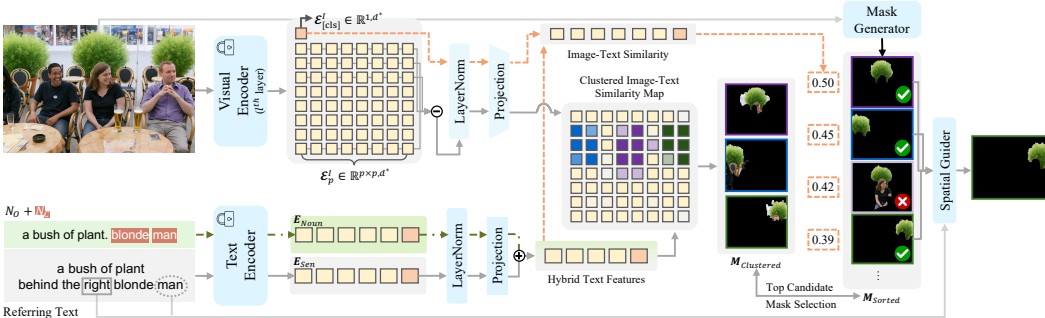

Figure 2: **The overall design of our COPATCH framework**. Our context-aware spatial map, CoMap, (*i.e.*, clustered image-text similarity map) is generated using: 1) Patch-level image features extracted from the $l$-th layer of the visual encoder. 2) Hybrid text features extracted with the combination of the global sentence and local primary noun and context token features. The spatial map is then used to select better top mask candidates, enabling precise final mask selection.

on minimal noun-phrase anchors or coarse spatial grounding signals (*e.g.*, GradCAM in Figure 1c), our design utilizes inherently embedded contextual tokens and spatial information from intermediate layers to enable richer spatial grounding at a lower cost.

### 3.1 CONTEXT TOKEN (CT) EXTRACTION FOR REFINED HYBRID TEXT FEATURES

**Motivation.** Previous studies have demonstrated the effectiveness of hybrid text features that combine global and local-level features. Herein, the local text feature encodes only the primary noun phrase. Specifically, it is commonly extracted as the primary noun phrase (Yu et al., 2023; Liu & Li, 2025) or chunk (Wang et al., 2025). However, this could leave out important spatial cues that might appear after the selected noun phrase (*e.g.*, "woman smiling" in "a bush of plant behind a middle woman smiling" in Figure 1a). Without appending contextual cues, different expressions may collapse to the same mask, limiting spatial disambiguation. Here, we hypothesize that augmenting context information when generating local-level features could improve spatial grounding.

**Method.** We construct hybrid text features that fuse both global and context-aware local representations. Given the referring expression $t$, we first extract global-level text features by feed-forwarding the entire text input into the text encoder ($\phi_\mathcal{T}$), resulting in $E_{\text{Sen}}(= \phi_\mathcal{T}(t))$. For local features, we extract two parts: primary noun phrase ($N_O$) and the context token ($N_C$) (implementation details in Appendix B.1). Then, these two chunks are concatenated at an input level and encoded jointly, generating $E_{\text{Noun}}(= \phi_\mathcal{T}([N_O|N_C]))$. Lastly, we merge the global-level and local-level text features by addition per hidden dimension ($E_{\text{ConText}} = \gamma E_{\text{Sen}} + (1 - \gamma)E_{\text{Noun}} \in \mathbb{R}^d$). Our final hybrid text feature is then used for generating our context-aware spatial map (similar attention-level merging in Appendix C.8). Unlike previous hybrid feature approaches that restrict local features to a single noun phrase, our formulation guides the model with better spatial cues from the contextual tokens for localizing target objects. This yields a more discriminative textual representation, serving as the foundation for our context-aware spatial map introduced in the next subsection.

### 3.2 CONTEXT-AWARE SPATIAL MAP FOR TOP CANDIDATE (TC) MASK SELECTION

**Motivation.** Although CLIP is often described as spatially blind (Tong et al., 2024; Tang et al., 2023; Kamath et al., 2023; Du et al., 2024; Yuksekgonul et al., 2023; Yu et al., 2023), we claim that this perception largely stems from their use of the original image extraction pathway. In CLIP, the final image feature is extracted as a single global vector at its final layer for image-text alignment, rendering the model inherently spatially agnostic. Our analysis reveals that its intermediate layers retain valuable, discriminative spatial information. Figure 3 illustrates that the cosine similarity between position-shifted images decreases with increasing layer depth, rising steeply at the final layer (unlike image-only feature extractor, DINO (Zhang et al., 2023; Tong et al., 2024)). This

observation motivates us to explicitly leverage intermediate patch embeddings, rather than using final-layer image features, for constructing our own spatial map.

**Method – Stage 1: Context-Aware Spatial Map Generation.** To better capture spatial cues, we compute image–text similarity at the patch level. Given the image $v$, we extract patch-level image embeddings using the *l-th exit* layer of the visual encoder ($\phi_\mathcal{V}^l$). At the *l-th* layer, we forward the patch-level image embeddings, $\mathcal{E}_\mathrm{p}^l \in \mathbb{R}^{p \times p \times d^*}$ into the post layer norm (LN). These embeddings are normalized and projected into the joint embedding space using the pretrained projection layer, resulting in the following patch-level image features: $E_\mathrm{p}^l = \mathrm{LN}(\mathcal{E}_\mathrm{p}^l) \cdot \mathbb{W}_{d^* \to d} \in \mathbb{R}^{p \times p \times d}$.

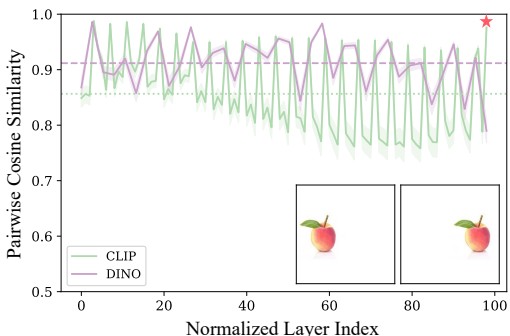

Figure 3: **Cosine similarity trend across layers for image pairs containing objects at different locations**. An overall decreasing trend is observed with a high peak (~1.0) at the last layer (indicated by $\star$) for CLIP.

We then compute the patch-level similarity map as follows: $\tilde{\mathcal{M}}^l = (\tilde{\mathcal{M}}_{ij}^l)_{1 \le i, j \le p} \in \mathbb{R}^{p \times p}$, where $\tilde{\mathcal{M}}_{ij}^l = \cos(E_\mathrm{p}^l[i,j], E_\mathrm{ConText}), \forall i, j \in \{1, ..., p\}$. This map highlights how strongly each image patch aligns with the referring expression, offering a spatially-enhanced alternative to the original CLIP's image features. Based on a property of cosine similarity, we present a simple method that resolves the well-known "opposite visualization" issue (see Appendix B.2) previously noted in Li et al. (2023) that appears in $\tilde{\mathcal{M}}^l$. We negate the patch embeddings before normalization as follows: $\mathcal{M}_{ij}^l = \cos(\tilde{E}_\mathrm{p}^l[i,j], E_\mathrm{ConText}^\top)$, where $\tilde{E}_\mathrm{p}^l[i,j] = \mathrm{LN}(-\mathcal{E}_\mathrm{p}^l[i,j]) \cdot \mathbb{W}_{d^* \to d}$. Unlike prior methods that rely solely on CLIP's global representation, this step directly restores spatial sensitivity from intermediate layers.

**Method – Stage 2: Mask Selection via Clustered Context-Aware Spatial Map (CoMap).** We present a framework to select the final mask out of mask candidates ($M$'s) generated from pretrained segmentation models, utilizing $\mathcal{M}^l$. To convert the patch-level activations into coherent spatial cues, we cluster adjacent patches with high similarity (Figure 4). Adjacent patches with high similarity form the same clusters, which are then interpolated into the final context-aware spatial map, CoMap.

The map is used to rerank the top candidate masks ($M_\mathrm{Sorted}$ in Figure 2) selected based on the original image-text similarity: $\mathcal{S}^m = \cos(E_\mathrm{Img}^m, E_\mathrm{ConText}^\top) \in \mathbb{R}, \forall m = 1, ..., M$ ($M$: # of predicted masks per sample). Specifically, we sort candidate masks ($M_\mathrm{Clustered}$ in Figure 2) based on their degree of overlap with all generated clusters from CoMap. Then, $M_\mathrm{Clustered}$ is used to filter out $M_\mathrm{Sorted}$ by 1) leaving the first candidate mask from $M_\mathrm{Sorted}$ and 2) reranking the top-$(k-1)$ candidate masks from $M_\mathrm{Sorted}$ based on the overlap with $M_\mathrm{Clustered}$. Among these top-$k$ candidate masks, the final mask is selected to be compared with the ground-truth mask (top-$k$ performance is also reported to validate the top-$k$ candidate masks). To ensure spatially consistent feature selection, we choose candidates based on the similarity between our hybrid text features and negative text features (*i.e.*,

---

**Algorithm 1** Spatial Map Clustering

**Require:** Spatial map $\mathcal{M} \in \mathbb{R}^{p \times p}$
1: $\mathcal{B} = (\mathcal{B}_{ij})$, where $\forall i, j: \mathcal{B}_{ij} \leftarrow [\mathcal{M}_{ij} > \delta]$
2: $\mathcal{C} \leftarrow \mathbf{0}_{p \times p}, l \leftarrow 0$
3: **for** each patch coordinate $\mathbf{k}$ in $\mathcal{B}$ **do**
4:     **if** $\mathcal{B}[\mathbf{k}] = 1$ **and** $\mathcal{C}[\mathbf{k}] = 0$ **then**
5:         $l \leftarrow l + 1$
6:         $Q \leftarrow$ new Queue()
7:         $Q.\mathrm{push}(\mathbf{k}), \mathcal{C}[\mathbf{k}] \leftarrow l$
8:         **while** $Q$ is not empty **do**
9:             $\mathbf{p} \leftarrow Q.\mathrm{pop}()$
10:             **for** each neighbor $\mathbf{n}$ of $\mathbf{p}$ **do**
11:                 **if** $\mathbf{n}$ is valid **and** $\mathcal{B}[\mathbf{n}] = 1$ **and** $\mathcal{C}[\mathbf{n}] = 0$ **then**
12:                     $Q.\mathrm{push}(\mathbf{n}), \mathcal{C}[\mathbf{n}] \leftarrow l$
13:                 **end if**
14:             **end for**
15:         **end while**
16:     **end if**
17: **end for**
18: **return** Interpolated $\mathcal{C}$

---

Figure 4: **Clustering algorithm for CoMap generation.** Clusters of the spatial map are formed by threshold-exceeding adjacent patches and interpolated into the final spatial map, CoMap (or $\mathcal{C}$).

| | Pretrained Models | RefCOCOg | | | RefCOCO | | | RefCOCO+ | | |
|---|---|---|---|---|---|---|---|---|---|---|
| | | val (U) | test (U) | val (G) | val | testA | testB | val | testA | testB |
| *Zero-shot RIS methods* **(CLIP ViT-B/32)** | | | | | | | | | | |
| Region Token (Suo et al., 2023; Li et al., 2022a) | - | 16.33 | 16.88 | 17.31 | 17.06 | 18.02 | 16.28 | 18.83 | 20.31 | 17.78 |
| CLIPSurgery (Li et al., 2023) | - | 20.44 | 21.80 | 21.23 | 18.04 | 14.34 | 21.28 | 18.39 | 14.34 | 22.98 |
| Cropping (Yu et al., 2023) | FreeSOLO | 31.88 | 30.94 | 31.06 | 24.83 | 22.58 | 25.72 | 26.33 | 24.06 | 26.46 |
| Global-Local (Yu et al., 2023) | FreeSOLO | 33.52 | 33.67 | 33.61 | 26.20 | 24.94 | 26.56 | 27.80 | 25.64 | 27.84 |
| Global-Local† | FreeSOLO | 32.23 | 33.11 | 31.80 | 23.90 | 21.74 | 25.85 | 24.18 | 21.97 | 26.45 |
| Global-Local (Yu et al., 2023; Suo et al., 2023) | SAM | 42.02 | 42.02 | 42.67 | 32.93 | 34.93 | 30.09 | 38.37 | 42.05 | 32.65 |
| HybridGL† (Liu & Li, 2025) | SAM | 44.52 | 44.93 | 44.52 | 38.26 | 39.43 | 37.91 | 36.04 | 38.65 | 32.07 |
| TAS (Suo et al., 2023) | SAM, BLIP-2 | 46.62 | 46.80 | 48.05 | 39.84 | 41.08 | 46.24 | 43.63 | 49.13 | 36.54 |
| COPATCH | SAM | 50.59 | 50.64 | 51.27 | 47.59 | 50.25 | 44.04 | 50.54 | 53.42 | 45.98 |
| COPATCH | Mask2Former | 52.24 | 52.15 | 51.69 | 46.76 | 52.09 | 40.18 | 44.07 | 50.66 | 35.86 |
| COPATCH (Top-3) | Mask2Former | 68.52 | 68.19 | 68.73 | 62.25 | 66.20 | 59.51 | 65.64 | 69.71 | 61.40 |
| Mask2Former Upper Bound† | - | 76.85 | 77.56 | 76.85 | 79.73 | 81.35 | 76.49 | 79.75 | 81.35 | 76.76 |
| *Zero-shot RIS methods* **(CLIP ViT-B/16)** | | | | | | | | | | |
| CaR (Sun et al., 2024) | SAM | 36.67 | 36.57 | 36.63 | 33.57 | 35.36 | 30.51 | 34.22 | 36.03 | 31.02 |
| Ref-Diff (Ni et al., 2023) | SAM, SD | 44.02 | 44.51 | 44.26 | 37.21 | 38.40 | 37.19 | 37.29 | 40.51 | 33.01 |
| HybridGL (Ni et al., 2023) | SAM | 51.25 | 51.59 | - | 49.48 | 53.37 | 45.19 | 43.40 | 49.13 | 37.17 |
| COPATCH | SAM | 57.67 | 57.88 | 58.35 | 50.06 | 58.94 | 50.88 | 51.65 | 50.75 | 52.93 |
| COPATCH | Mask2Former | 54.42 | 55.62 | 54.71 | 50.05 | 55.38 | 43.01 | 46.83 | 54.08 | 37.67 |
| COPATCH (Top-3) | Mask2Former | 71.21 | 72.29 | 72.11 | 65.38 | 67.77 | 62.45 | 67.41 | 69.86 | 63.83 |
| *Zero-shot RIS methods* **(DFN ViT-H/14)** | | | | | | | | | | |
| COPATCH | Mask2Former | 56.23 | 55.88 | 55.68 | 50.91 | 56.59 | 44.79 | 47.05 | 54.12 | 38.91 |
| COPATCH (Top-3) | Mask2Former | 74.23 | 74.90 | 75.04 | 68.33 | 72.00 | 65.55 | 71.26 | 74.22 | 68.04 |
| *Weakly/Pseudo-supervised RIS methods* | | | | | | | | | | |
| CHUNK (ALBEF) (Lee et al., 2023) | - | 32.90 | - | - | 31.10 | 32.30 | 30.10 | 31.30 | 32.10 | 30.10 |
| TSEG (ViT-S/16 & BERT) (Strudel et al., 2022) | - | 23.41 | - | - | 25.95 | - | - | 22.62 | - | - |
| PPT (CLIP ViT-B/16) (Dai & Yang, 2024) | SAM | 42.97 | - | - | 46.76 | 45.33 | 46.28 | 45.34 | 45.84 | 44.77 |
| Pseudo-RIS (ResNet50) (Yu et al., 2024) | SAM, CoCa | 45.99 | 46.67 | 46.80 | 41.05 | 48.19 | 33.48 | 44.33 | 51.42 | 35.08 |

Table 1: **Comparison of mIoU performances across zero-shot and weakly/pseudo-supervised RIS methods on RefCOCOg, RefCOCO, and RefCOCO+ datasets**. We attain the overall best performances for CLIP (ViT-B/32 and ViT-B/16) and DFN (ViT-H/14), outperforming the SoTA, TAS (Suo et al., 2023), and HybridGL (Liu & Li, 2025). † stands for the reproduced results. The **bolded** and underlined texts indicate the best and the second-best models, excluding the models with top-3 and the upper bound performance.

features of primary noun(s) that come after the positional relation) if an explicit spatial cue (*e.g.*, 'behind') is present leveraging the spatial guider (*i.e.*, Spatial Coherence or SC from Liu & Li (2025)).

Our approach challenges the prevailing assumption that CLIP is spatially blind by unlocking latent spatial knowledge from its intermediate representations. In contrast to prior methods that rely on computationally intensive gradient techniques or attention maps, our approach directly extracts spatially-aware features using our CoMap. This training-free extraction without recomputation is not only more efficient but also more faithful to the model's inherent spatial understanding.

## 4 EXPERIMENTS

### 4.1 EXPERIMENTAL SETTINGS

The details on the datasets and evaluation metrics are in Appendix A. For implementation, we use CLIP (ViT-B/32 and ViT-B/16) (Radford et al., 2021) as a model backbone to ensure a fair comparison with previous zero-shot RIS methods, as well as DFN (ViT-H/14) (Fang

| | Train Dataset | *All* | *Unseen* |
|---|---|---|---|
| *Zero-shot RIS methods* | | | |
| Global-Local (Yu et al., 2023) | ✗ | 23.64 | 22.98 |
| Global-Local† | ✗ | 22.43 | 21.54 |
| TAS (Suo et al., 2023) | ✗ | 25.64 | 24.66 |
| Ref-Diff (Ni et al., 2023) | ✗ | 29.42 | - |
| IteRPrimE (Wang et al., 2025) | ✗ | 38.10 | 37.90 |
| HybridGL (Liu & Li, 2025) | ✗ | 38.39 | - |
| HybridGL† | ✗ | 37.60 | 37.91 |
| COPATCH (Top-1) | ✗ | **38.53** | **38.31** |
| COPATCH (Top-3) | ✗ | 45.36 | 45.14 |
| *Supervised RIS methods* | | | |
| CRIS (Wang et al., 2022b) | RefCOCO | 15.53 | 13.75 |
| | RefCOCO+ | 16.30 | 14.62 |
| | RefCOCOg | 16.24 | 13.88 |
| LAVT (Yang et al., 2022) | RefCOCO | 16.68 | 14.43 |
| | RefCOCO+ | 16.64 | 13.49 |
| | RefCOCOg | 16.05 | 13.48 |
| Pseudo-RIS (Yu et al., 2024) | RefCOCO+ | 28.68 | 29.14 |
| | PhraseCut | 32.75 | 33.52 |

Table 2: **Comparison of oIoU performances across zero-shot and supervised RIS methods on PhraseCut dataset**. COPATCH (CLIP ViT-B/16) surpasses all the existing methods, including Pseudo-RIS (Yu et al., 2024) trained on PhraseCut datasets.

et al., 2023) that shows a better spatial grounding capability among CLIP variations (Tong et al., 2024). Before the final evaluation, we perform a hyperparameter search to find the optimal exit layer ($l$) of the visual encoder, the initial threshold ($\delta$), and the proportion of spatial coherence ($\alpha$)

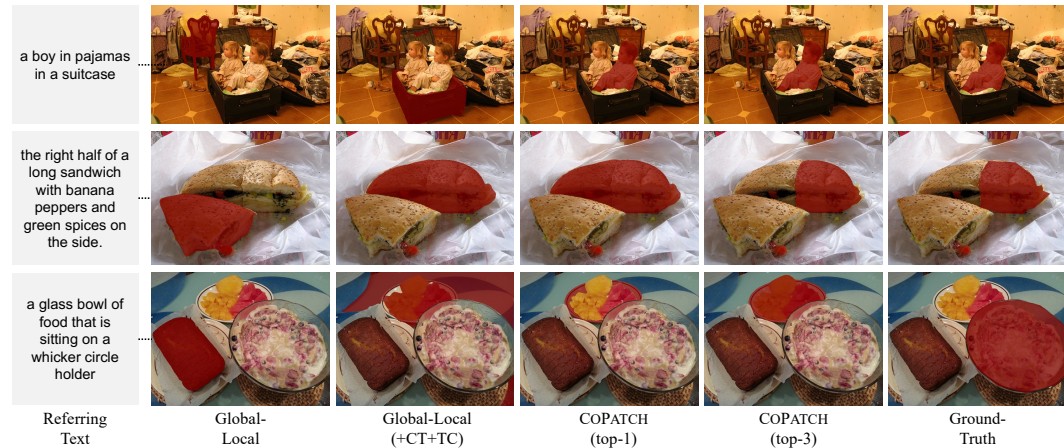

| Referring Text | Global-Local | Global-Local (+CT+TC) | COPATCH (top-1) | COPATCH (top-3) | Ground-Truth |

Figure 5: **Qualitative comparison across zero-shot RIS methods**. Our COPATCH demonstrates strong spatial grounding ability. The first row shows a case where both COPATCH top-1 and top-3 methods segment correctly compared to Global-Local (Yu et al., 2023) and new Global-Local (+CT+TC, as in Table 3a). The second row captures a case that is incorrectly segmented using COPATCH (top-1) but accurately segmented on the "right half" with COPATCH (top-3).

(Section 3.2). This is conducted using only a 10% of the RefCOCOg val set, and we use fixed $l$, $\delta$, and $\alpha$ for all evaluations ($l$: 10, 8, and 22, $\delta$: 0.5, 0.3, 0.5, and $\alpha$: 0.5, 0.7, and 0.5 for COPATCH (CLIP ViT-B/32, CLIP ViT-B/16, and DFN ViT-H/14).

## 4.2 MAIN RESULTS

**Comparison to State of the Arts.** As shown in Table 1 (more results in Appendices C.3 – oIoU performance and C.10 – different VLM architectures) and Figure 5, we achieve notable superior performance over existing SoTA zero-shot methods (Yu et al., 2023; Suo et al., 2023; Liu & Li, 2025; Ni et al., 2023; Li et al., 2023; Sun et al., 2024), even higher than the results of weakly- and pseudo-supervised approaches (Strudel et al., 2022; Lee et al., 2023; Yu et al., 2024) (the results using different pre-trained models in Appendix C.2). We also note that using Mask2Former achieves lower infer-

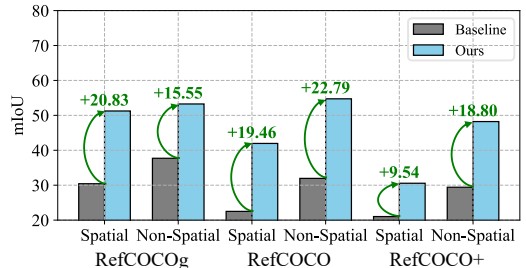

Figure 6: **Subset performance comparison between Global-Local (Yu et al., 2023) and CO-PATCH**. We observe a significant gain for all the spatial and non-spatial validation subsets.

ence cost than SAM (*e.g.*, ×5 per dataset) while performing still the best in both CLIP ViT-B/32 and ViT-B/16. We also test the validity of our method on DFN (Fang et al., 2023), where it attains the best results on 8 out of 9 evaluation splits. COPATCH with CLIP backbones also demonstrates strong performance and is enhanced further if the final mask is selected among the top-3 candidate masks generated using our `CoMap`. For example, in the RefCOCOg validation (U) and test sets with ViT-B/32, we observe gains of +16.28 and +16.04 mIoU, respectively, using Mask2Former. These results reach 89.16% and 87.91% of the mask upper bound (oracle) performances. Since this oracle performance indicates the IoU between the *best* mask generated by the pretrained segmentation model and the ground-truth mask, the closer gap between ours and oracle performance suggests that the top candidate masks selected using the clusters from `CoMap` are highly informative (than previous methods) in locating the target object (qualitative results shown in Figure 5, second row). Although there are still several cases where COPATCH may not be able to segment correctly due to CLIP's limited understanding of some vocabularies (*e.g.*, "glass" in Figure 5, third row), our approach achieves the best results even on the very challenging PhraseCut dataset. We obtain oIoU scores of 38.41 (*all*) and 38.12 (*unseen*), outperforming recent SoTA zero-shot and supervised methods (Table 2). These results highlight the robustness and generalizability of our method across various datasets.

**Spatial Subset Evaluation.** To evaluate the fine-grained spatial grounding ability of CLIP, we conduct a targeted evaluation on samples that contain explicit spatial cues (*e.g.*, "left", "bottom", and "closest to"). To identify such samples, we utilize Qwen2.5-14B-Instruct (Yang et al., 2024) to classify each referring expression as spatial or non-spatial and extract the cue if present (prompt details in Appendix C.6). These automatically generated annotations take up 50.72%, 62.40%, and 23.48% of the validation samples in RefCOCOg, RefCOCO, and RefCOCO+. Figure 6 shows significant gains using COPATCH over the baseline, with +20.83% and +15.55% mIoU improvements on the spatial and non-spatial subsets of RefCOCOg. These improvements demonstrate that our spatial map `CoMap` contributes to the gains on spatial subsets, while the integration of context-aware tokens enhances performance on non-spatial subsets by providing richer semantic information.

## 4.3 ABLATION STUDIES AND ANALYSIS

**Spatial Map Analysis.** We analyze the effectiveness of our `CoMap` by comparing it with the widely-used Grad-CAM (Selvaraju et al., 2016) used in SoTA zero-shot RIS method with ALBEF (Li et al., 2021) as backbone, IteR-PrimE (Wang et al., 2025). Figure 7 illustrates a qualitative comparison between Grad-CAM (used in IteRPrimE) and `CoMap` used in our method. We observe that `CoMap` better captures target objects in many cases (*e.g.*, "chair" in the first column). Moreover, `CoMap` (ViT-B/16) achieves better average performance (*e.g.*, +6.9 mIoU) with lower inference latency (*e.g.*, −2.95 *sec*. per sample), as shown in the Appendix C.1.

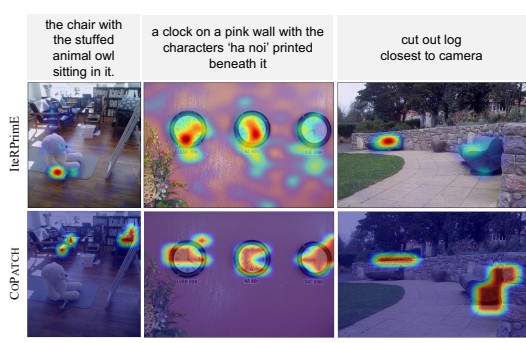

Figure 7: **Spatial map comparison of Grad-CAM used in IteRPrimE and `CoMap` used in our COPATCH**. Our spatial map better localizes the target objects (*e.g.*, "chair", "clock", "log") in referring expressions.

In addition, we evaluate how our spatial map contributes to the quality of top candidate masks. Unlike prior methods (Yu et al., 2023) that rely on global image-text cosine similarity, our map captures fine-grained patch-level spatial correspondences. This is particularly important in scenarios where distinguishing targets according to their spatial configuration is essential for accurate localization. To reflect this challenge, we design our experiments to select the final mask given different numbers of top candidate masks. Table 3a shows that our approach consistently outperforms Global-Local, demonstrating the effectiveness of our `CoMap` for generating accurate and reliable region proposals (qualitative results in Figure 8). Specifically, we select top-$k$ candidate masks based on the ranking of original image-text similarity for Global-Local and new image-text similarity reranked using the automatically generated clusters from `CoMap` for COPATCH. Using the top three candidate masks consistently yields the best performance (Table 3a) and sufficient diversity (Figure 8) for the final mask selection.

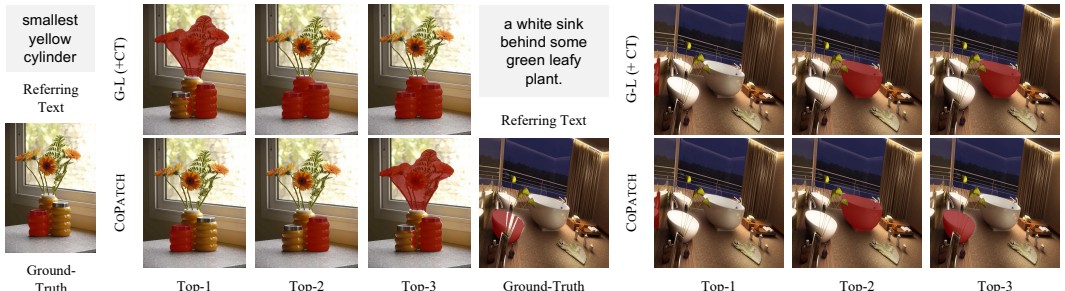

Figure 8: **Qualitative comparison of top candidate masks generated using original image-text similarity from Global-Local (w/ CT) and averaged `CoMap` similarity score using our CO-PATCH**. The top candidate masks generated using our method show better diversity and accuracy.

(a) **Comparison of zero-shot RIS performances using top candidates selected by the original and our candidate mask scores on RefCOCOg val (U) set.**

| Top Candidate Mask Number | Global-Local mIoU | Global-Local oIoU | CoPatch mIoU | CoPatch oIoU |
|---|---|---|---|---|
| 2 | 50.47 | 40.28 | **52.08** | **40.72** |
| 3 | 50.79 | 41.29 | **52.24** | **41.43** |
| 4 | 49.42 | 40.81 | **51.92** | **41.40** |
| $k$ | – | – | **51.80** | **41.43** |

(b) **Comparison of zero-shot RIS performances without and with each subcomponent.**

| | CT | TC | SC | ViT-B/32 mIoU | ViT-B/32 oIoU | ViT-B/16 mIoU | ViT-B/16 oIoU |
|---|---|---|---|---|---|---|---|
| HybridGL | | | | 41.54 | 29.89 | 46.59 | 37.04 |
| CoPatch | ✓ | | | **42.58** | **30.77** | **47.01** | **37.36** |
| Global-Local | | | | 47.07 | 35.73 | 50.30 | 38.35 |
| CoPatch | ✓ | | | **49.65** | **37.18** | **50.65** | 38.24 |
| Global-Local | ✓ | ✓ | | 47.70 | 36.32 | 50.68 | **38.84** |
| CoPatch | ✓ | ✓ | | **50.16** | **37.63** | **51.16** | **38.84** |
| Global-Local | ✓ | ✓ | ✓ | 50.42 | 38.17 | 50.91 | 39.58 |
| CoPatch | ✓ | ✓ | ✓ | **52.24** | **41.43** | **54.42** | **44.14** |

Table 3: **Ablation study results using different numbers of top mask candidates and subcomponents**. (a) Our performances are consistently greater than when using top mask candidates selected by Global-Local (Yu et al., 2023) ($k$: # of automatically generated clusters that differ per data sample). (b) Using the baseline feature extraction methods–Global-Local (Yu et al., 2023) and HybridGL (Liu & Li, 2025), all the subcomponents, Context Token (CT), Top Candidate (TC), and Spatial Coherence (SC) are integral for building CoPatch.

| | Negation? | Clustering? | RefCOCOg val (U) | RefCOCOg test (U) | RefCOCOg val (G) | RefCOCO val | RefCOCO testA | RefCOCO testB | RefCOCO+ val | RefCOCO+ testA | RefCOCO+ testB |
|---|---|---|---|---|---|---|---|---|---|---|---|
| *Zero-shot RIS methods* (CLIP ViT-B/16) - mIoU | | | | | | | | | | | |
| CoPatch | | ✓ | 49.53 | 50.67 | 49.41 | 40.15 | 44.52 | 35.56 | 41.99 | 46.38 | 35.36 |
| CoPatch | ✓ | | 50.00 | 51.16 | 49.91 | 41.52 | 47.45 | 35.23 | 42.45 | 47.29 | 34.77 |
| CoPatch | ✓ | ✓ | **54.42** | **55.62** | **54.71** | **50.05** | **55.38** | **43.01** | **46.83** | **54.08** | **37.67** |
| *Zero-shot RIS methods* (CLIP ViT-B/16) - oIoU | | | | | | | | | | | |
| CoPatch | | ✓ | 40.53 | 42.34 | 41.71 | 33.92 | 40.06 | 31.69 | 34.53 | 40.12 | 31.01 |
| CoPatch | ✓ | | 39.86 | 42.48 | 41.13 | 34.41 | 41.12 | 30.43 | 34.39 | 39.63 | 29.57 |
| CoPatch | ✓ | ✓ | **44.13** | **46.24** | **45.26** | **40.85** | **48.45** | **34.98** | **38.39** | **46.17** | **32.15** |

Table 4: **The effect of negation and clustering on our COMAP generation.** Omitting negation or clustering reduces both mIoU and oIoU performance, demonstrating the importance of both components for generating intuitive similarity maps.

**Subcomponents.** Table 3b demonstrates the effectiveness of the core components of Co-Patch – Context Token (CT) extraction (Section 3.1) and Top Candidate (TC) mask selection (Section 3.2). We use Global-Local (Yu et al., 2023) as the backbone image/text extraction method since the vanilla performances are higher than those of HybridGL (Liu & Li, 2025) (*e.g.*, 5.53 higher mIoU on ViT-B/32). We find that using our CT alone helps the models to achieve higher performances, especially

| Category | Punct? | Count | Mean IoU |
|---|---|---|---|
| Full sentence | | 26 | 0.658 |
| Fragmented sentence | ✓ | 60 | 0.578 |
| Full sentence | ✓ | 31 | 0.559 |
| Fragmented sentence | | 68 | 0.513 |

Figure 9: **Performance of CoPatch (CLIP ViT-B/16) across different grammatical categories.** The performance is varies across different types of referring expressions.

in ViT-B/32 (*e.g.*+2.58 for mIoU; qualitative results in Appendix C.5). After the text feature extraction, mask scoring with top candidate masks from our ranked clusters (*i.e.*, TC) proves more effective than the previous approach (Liu & Li, 2025), which ranks masks solely by the cosine similarity between raw image and text features. Specifically, our method achieves improvements of +2.46 and +1.32 mIoU and oIoU on ViT-B/32. Finally, the mIoU and oIoU performances improve by 2.80% and 4.18% using Spatial Coherence (SC) (Liu & Li, 2025).

In addition to these three components, we analyze the effect of integral steps of CoMap generation. As can be observed in Table 4, we find that the negation and clustering strategies are integral since the performance drops by 4–11% if either of these steps is not performed. On top of that, we observe that while CT extraction is crucial, the overall performance is still sensitive to fragmented referring expressions (Table 9). Full sentences without punctuation may perform better than those with punctuation because lengthy prompts often contain punctuations, and overly specific, long phrases can make it harder for the model to comprehend (*e.g.*, +2 words on average in our experiment). To sum up, all these results confirm that each component of CoPatch contributes meaningfully, and their synergy is crucial for achieving strong zero-shot RIS performance.

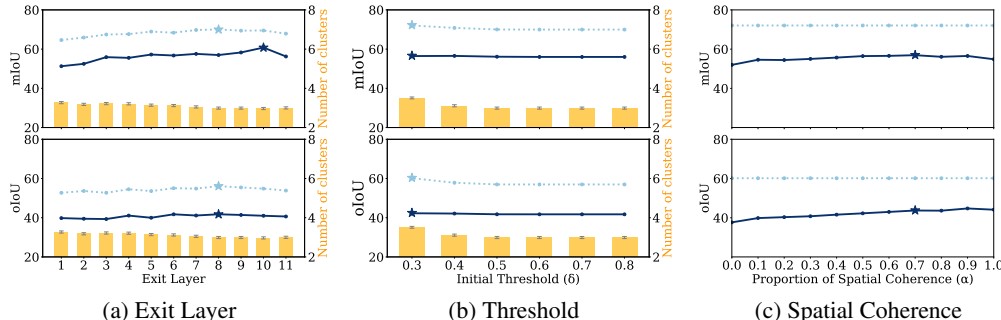

(a) Exit Layer  (b) Threshold  (c) Spatial Coherence

Figure 10: **Hyperparameter tuning results for CO PATCH (ViT-B/16) using 10% of the Ref-COCOg val set**. The **deep-blue** and **sky-blue** lines depict **top-1** and **top-3** performances across exit layers. The **golden** bars represent the number of clusters averaged across data instances for each exit layer. The overall IoU performance is insensitive to a particular hyperparameter value.

**Hyperparameters.** Figure 10 shows the performance change across different exit layers (Appendix C.9), initial threshold, and proportion of the spatial coherence method (Liu & Li, 2025). We observe no significant drop in performance across varying hyperparameters tuned on 10% of the RefCOCOg validation set (Mao et al., 2016). For all experiments in the main paper (Tables 1 and 2), we use a single fixed hyperparameter configuration (Appendix C.7). Despite this global setting, CO PATCH consistently outperforms previous methods across benchmarks, demonstrating that our method is robust and generalizes well without requiring dataset-specific tuning. However, more optimal performances could also be achieved if hyperparameter tuning is performed.

## 5 CONCLUSION

We propose CO PATCH, a novel zero-shot RIS framework that addresses the spatial limitations of CLIP by improving both its text and image representations. On the text side, CO PATCH enriches linguistic understanding by incorporating spatially informative context tokens, including beyond primary noun phrases commonly used in prior works. On the image side, we exploit spatial information preserved in CLIP's intermediate layers to generate fine-grained patch-level image features. These are used to compute the context-aware spatial map, `CoMap`, which enables accurate top mask selection. Experimental results demonstrate that CO PATCH consistently outperforms the SoTA zero-shot RIS methods across RefCOCO, RefCOCO+, RefCOCOg, and PhraseCut benchmarks, with 2–7% gains across different CLIP architectures. Notably, we achieve a 10–21% improvement on a challenging spatial subset. Our synergistic approach to enhancing both visual and textual representations opens promising directions for spatially-grounded, semantically-robust zero-shot segmentation.

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

APPENDIX

## A EXPERIMENTAL SETUPS

**Datasets.** We evaluate our method on widely used RIS benchmarks: RefCOCO (Nagaraja et al., 2016), RefCOCO+ (Nagaraja et al., 2016), and RefCOCOg (Mao et al., 2016). These datasets have diverse linguistic and visual structures, where RefCOCO and RefCOCO+ have various objects of the same category per image, and RefCOCOg has relatively longer, complex referring expressions (Yang et al., 2022). RefCOCO includes more explicit positional relations (*e.g.*, "left", "bottom") than RefCOCO+ and RefCOCOg. In addition, following previous works, we include evaluations on the PhraseCut dataset (Wu et al., 2020), which contains various phrase types such as attributes (*e.g.*, large), categories (*e.g.*, cake), and relationships (*e.g.*, on plate). PhraseCut is evaluated in two settings − *all* (samples) and *unseen*, where the latter indicates evaluation only on samples that do not contain a word from pre-defined 91 MS COCO (Lin et al., 2014) classes.

**Evaluation Metrics.** For evaluation metrics, we use Mean Intersection over Union (mIoU) and Overall Intersection over Union (oIoU), following related works. The mIoU is computed as the average IoU (= $\frac{|P \cap G|}{|P \cup G|}$, where $P$: predicted mask generated with a mask generator and $G$: ground-truth mask) across all data instances. The oIoU is computed as the total intersection over the total union across all samples.

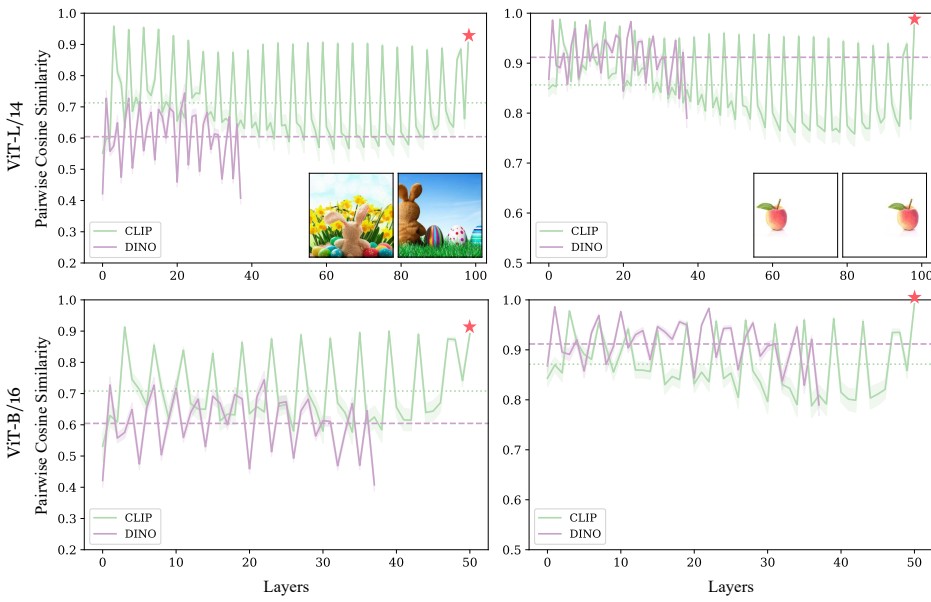

Figure 11: **Varying cosine similarity across layers for image pairs containing objects in different locations for different backbones and datasets**. We observe a decreasing overall trend of similarity for CLIP and a high peak at the last layer.

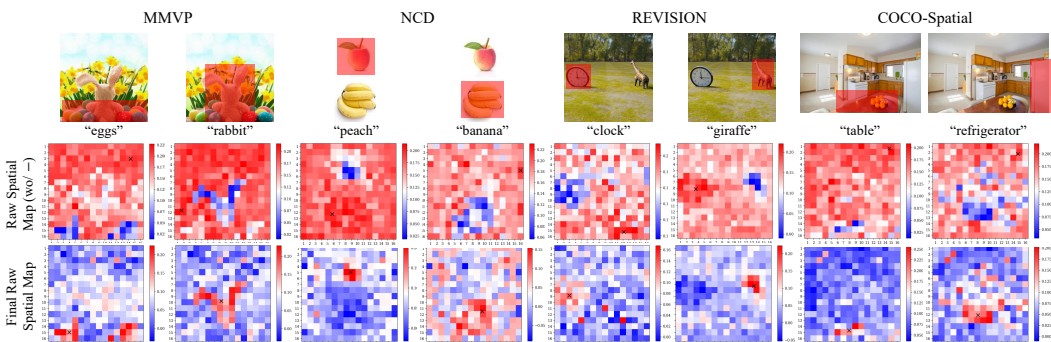

Figure 12: **Opposite visualization without the negation (above) and our final raw spatial map (bottom)**. The final raw spatial map localizes the "target objects" and resolves the opposite visualization issue.

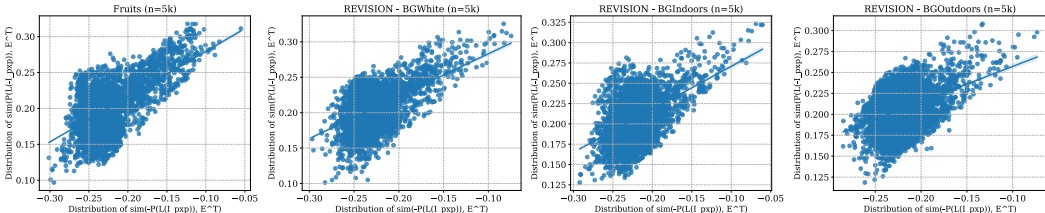

Figure 13: **Correlation plots between our final spatial map (`CoMap`) image-text similarity scores (y-axis) and negative of the image-text similarity scores from the oppositely visualized spatial map (x-axis)**. There exists a strong positive correlation between these two similarity score values across data samples and datasets.

## B    METHOD DETAILS

### B.1    CONTEXT TOKEN (CT) EXTRACTION FOR REFINED HYBRID TEXT FEATURES

Following previous work (Wang et al., 2025), we use the NLP software library, stanza (Qi et al., 2020), to automatically extract $N_O$ and $N_C$, where $N_C$ are noun or adjective tokens that are not contained in $N_O$ that may appear right side of the primary noun phrase/chunk. Note that we observe a slight performance improvement using stanza instead of spacy (Honnibal, 2017). For instance, using Stanza and Spacy on our method (CoPatch) in the RefCOCOg (val) dataset results in IoU performances of 52.24 and 51.50, respectively. Based on our preliminary analysis, we notice that this method is particularly beneficial when $N_C$ consists of noun chunks that are well-captured by the model (*e.g.*, "woman smiling") and color adjectives (qualitative results in Appendix C.4).

### B.2    NOVEL SPATIAL MAP FOR TOP CANDIDATE (TC) MASK SELECTION

**Method – Stage 1: Context-Aware Spatial Map Generation.**    We explain why the negation operation is necessary to create an intuitive spatial map (`CoMap`) that resolves the opposite visualization problem described in previous work (Li et al., 2023). To begin with, the spatial map is built based on the observation that the cosine similarity between embeddings of image pairs containing identical objects in different locations is lower in intermediate layers than that of the final layer (Figure 11). This decreasing trend phenomenon implies that the intermediate image embeddings may better preserve spatial information than the original image features extracted from the last layer of the visual encoder.

However, when we calculate the similarity between the hybrid text feature ($E_{\text{ConText}} \in \mathbb{R}^d$) and patch-level image features extracted from the $l - th$ exit layer ($E_p^l \in \mathbb{R}^{p \times p \times d}$), the similarity score values of the resulting similarity map ($\tilde{\mathcal{M}}^l \in \mathbb{R}^{p \times p}$) is shown with opposite visualization (Figure 12). If we simply negate the similarity map itself, the opposite visualization phenomenon is naturally resolved,

but the range of similarity scores falls within -0.3 to 0.0 ($x$-axis in Figure 13). We could shift these scores by an addition in order to make them fall into the original range of similarity scores (0.0 to 0.3). However, this postprocessing method requires an additional hyperparameter for the shifting and does not fundamentally solve the opposite visualization problem.

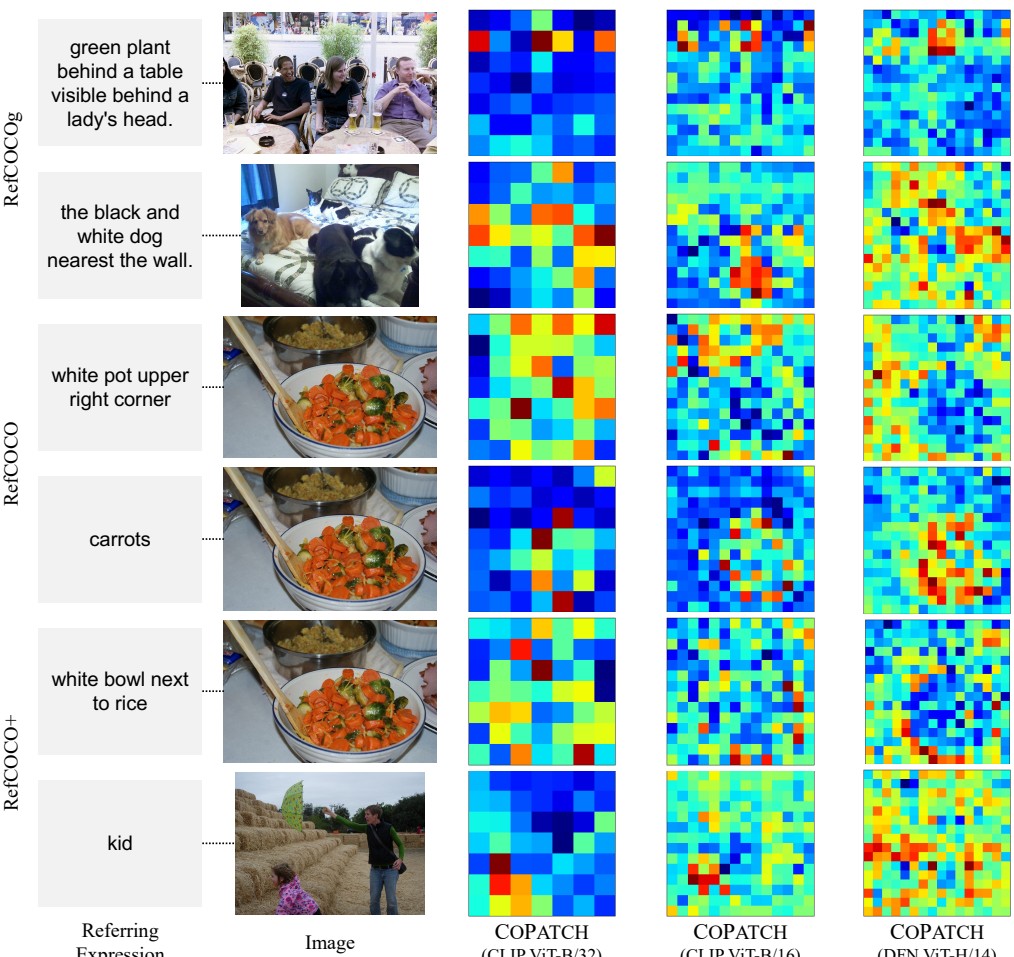

Figure 14: **Comparison of raw spatial maps using different backbones for COPATCH**. The spatial map becomes more fine-grained for the backbone with higher dimensions for the number of patches ($7 \times 7$ in CLIP ViT-B/32; $14 \times 14$ in CLIP ViT-B/16; $16 \times 16$ in DFN ViT-H/14).

Surprisingly, simply negating the patch-level image embeddings before feeding them to the post-layer norm and multiplying them with the projection weight naturally resolves the opposite visualization and out-of-range issues. As shown in Figure 13, the correlation between values of $\mathcal{M}^l_{ij} = \cos(\tilde{E}^l_p[i,j], E_{\text{ConText}})$ ($y$-axis) and $\mathcal{M}^l_{\text{Neg}} = -\cos(E^l_p[i,j], E_{\text{ConText}})$ ($x$-axis) is positive ($\forall i, j = \{1, ..., p\}$). This observation is a product of the definition of cosine similarity:

$$\cos(-x, y) = \frac{(-x) \cdot y}{\| -x \| \| y \|} = \frac{-(x \cdot y)}{\| x \| \| y \|} = -\cos(x, y)$$

This suggests that the feature negation can serve as a proxy for semantic opposition, providing an effective mechanism at a low cost for enhancing the model's spatial reasoning. Our raw spatial map ($\mathcal{M}^l = (\mathcal{M}^l_{ij})_{1 \le i,j \le p}$) can be found in Figure 14 across different backbones. The final interpolated, clustered spatial map, CoMap, using different exit layers can also be observed in Figure 15.

**Method – Stage 2: Mask Selection via Clustered Context-Aware Spatial Map (CoMap).** During the clustering process, our algorithm first creates a binary mask ($\mathcal{B}$) by thresholding the input

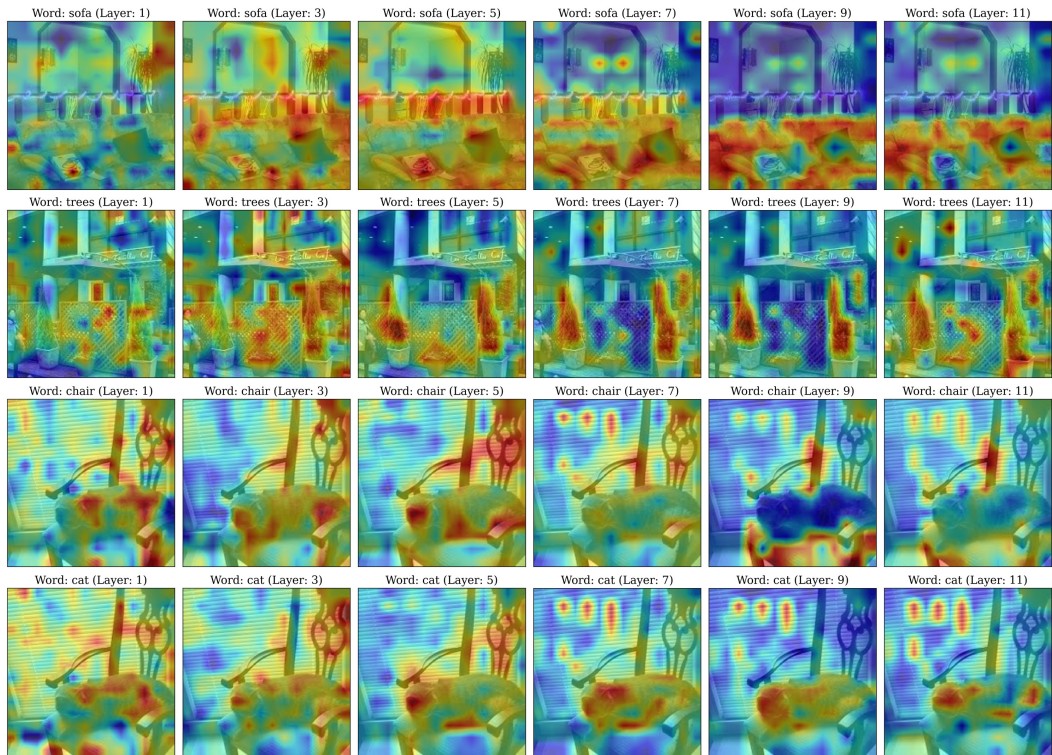

Figure 15: **Comparison of our spatial map, `CoMap` generated using varying exit layers**. Similar to the mIoU performance trend across different exit layers in Figure 9a, `CoMap` captures the most intuitive image-text similarity using the intermediate layers close the last layer (*e.g.*, layer 9).

continuous spatial map ($\mathcal{M}$) to identify all relevant patches (line 1). An output cluster map ($\mathcal{C}$) is initialized with all zeros, and a counter ($l$) is also initialized to 0 (line 2). Then, each patch coordinate in the binary mask is iterated to find and assign a label to an unlabeled, active patch, which serves as the seed for a new cluster (lines 3-7). The algorithm then initiates a Breadth-First Search (BFS) from the seed, which continues as long as the queue ($\mathcal{Q}$) is not empty, to progressively label all patches belonging to the same connected component. Note that the number of automatically generated clusters from our `CoMap` ($k$) may differ per sample.

## C  ADDITIONAL RESULTS

Here, we explain results not shown in the main paper. All inference is conducted on a single RTX A4000 (16 Gi) for all the models with CLIP ViT-B/32 and ViT-B/16 backbones. For the DFN ViT-H/14 backbone, a single RTX A6000 (48 Gi) is used. To ensure a fair comparison for calculating the inference time in Table 5, we use the same device, RTX A6000, for all the methods.

### C.1  SPATIAL MAP COMPARISON

**Segmentation Approaches.**  Our final spatial map, `CoMap`, is surprisingly comparable to several zero-shot segmentation models, such as CLIPSurgery (Li et al., 2023) (Figure 16) and CLIP-Trace (Shao et al., 2024) (Figure 17). CLIPSurgery is generated via recomputation of the self-attention operation ($A = \sigma(s \cdot VV^T)V$ instead of $A = \sigma(s \cdot QK^T)V$, where $s$: scaling factor) for every layer (before the start of the new path). Similarly, CLIPTrace also requires a recomputation of self-attention for producing a more consistent semantic correlation matrix. However, `CoMap` does not require recomputations for self-attention modules and produces intuitive activation patterns similar to those of these segmentation models.

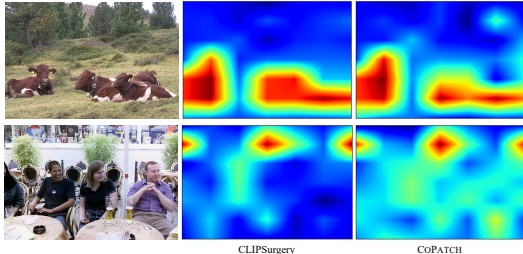

Figure 16: **Activation pattern comparison between CLIPSurgery and COPATCH**. Our spatial map shows a very similar activation as CLIPSurgery without attention recomputations.

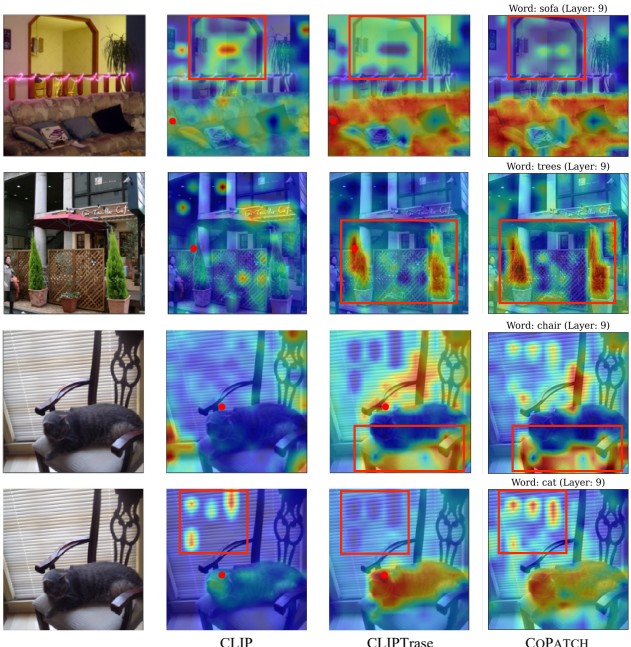

Figure 17: **Activation pattern comparison between CLIPTrace and COPATCH**. Although CO-PATCH does not show completely clean activations without noises (red blobs on the last row), it can still capture highly activated areas like CLIPTrace.

| | RefCOCOg | | RefCOCO | | RefCOCO+ | | Average | |
|---|---|---|---|---|---|---|---|---|
| | Time | mIoU | Time | mIoU | Time | mIoU | Time | mIoU |
| IterPRIMe Wang et al. (2025) | 3.20 | 46.0 | 5.42 | 40.2 | 4.82 | 44.2 | 4.48 | 43.5 |
| COPATCH (CLIP ViT-B/32) | **1.27** | 52.2 | **1.45** | 46.8 | **1.44** | 44.1 | **1.38** | 47.7 |
| COPATCH (CLIP ViT-B/16) | 1.40 | 54.4 | 1.60 | 50.1 | 1.60 | 46.8 | 1.53 | 50.4 |
| COPATCH (DFN ViT-H/14) | 2.24 | **56.2** | 2.52 | **50.9** | 2.53 | **47.1** | 2.43 | **51.4** |

Table 5: **Inference time and performance comparison between IteRPrimE and COPATCH on validation (U) sets**. We report the time in seconds per iteration, averaged across the first 100 samples for each dataset. Our methods comparatively take less time than IterPRIMe while achieving better IoU performance.

**Grad-CAM.** When compared to Grad-CAM (Selvaraju et al., 2016), which is often applied in several zero-shot RIS approaches (Subramanian et al., 2022; Wang et al., 2025; Lee et al., 2023), Figure 18 demonstrates that our CoMap can provide better spatial cues for a primary object from a referring expression. Furthermore, our COPATCH (using CoMap) takes less inference time than IteRPrimE, which uses Grad-CAM, as shown in Table 5 while achieving better IoU performance.

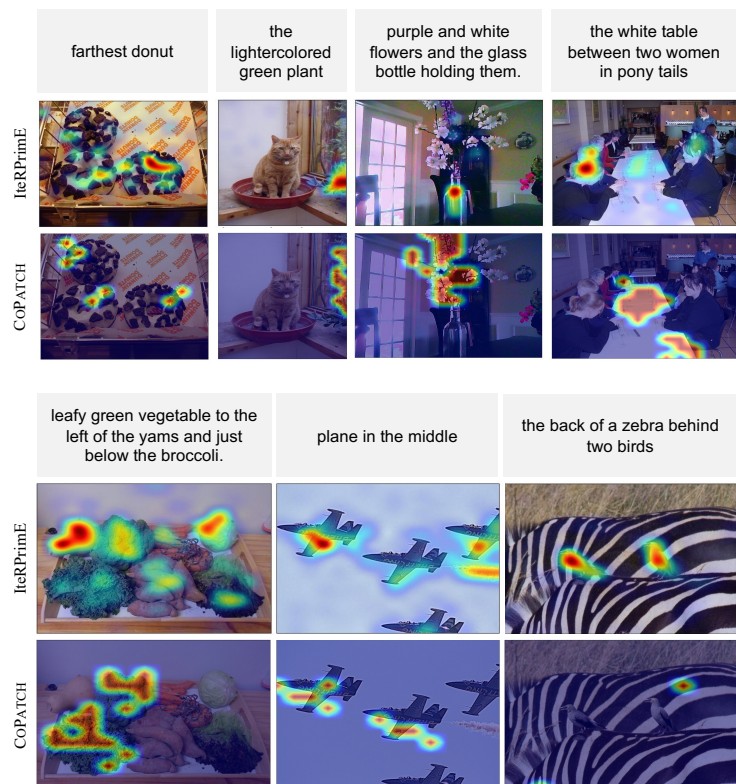

Figure 18: **Qualitative comparison between Grad-CAM (IteRPrimE) and `CoMap` (COPATCH).** Our `CoMap` can better capture and localize the primary objects in the referring expressions.

|  | FreeSOLO | | SAM | | Mask2Former | |
|---|---|---|---|---|---|---|
|  | mIoU | oIoU | mIoU | oIoU | mIoU | oIoU |
| Global-Local (CLIP ViT-B/32) Yu et al. (2023) | 33.59 | 31.02 | 38.56 | 27.04 | **45.23** | **32.59** |
| HybridGL (CLIP ViT-B/32) Liu & Li (2025) | 30.66 | 27.97 | 41.54 | 29.89 | **44.27** | **32.17** |
| HybridGL† (CLIP ViT-B/16) | 30.84 | 28.02 | **46.59** | **37.04** | 43.90 | 31.73 |
| IteRPrimE (ALBEF) Wang et al. (2025) | 29.03 | 26.36 | 25.23 | 21.27 | **46.89** | **41.09** |
| Mask Upper Bound (CLIP ViT-B/32) | 52.44 | 52.76 | 74.39 | 75.76 | 76.85 | 72.73 |

Table 6: **Zero-shot performances of SoTA referring image extraction methods applied with three mask generators on RefCOCOg val (U) set**. The highest IoU performances are generally achieved using Mask2Former across different methods. † stands for the reproduced results.

## C.2 MASK GENERATOR SELECTION

For selecting the mask generator, we compare three widely used pretrained segmentation models, FreeSOLO (Wang et al., 2022a), SAM (Kirillov et al., 2023), and Mask2Former (Cheng et al., 2022; Liang et al., 2023) on several SoTA zero-shot RIS approaches (Yu et al., 2023; Wang et al., 2025; Liu & Li, 2025). As can be observed in Table 6, we observe that Mask2Former is most effective in the RefCOCOg validation (U) set (Mao et al., 2016). Compared to SAM, it is also highly efficient, taking a fifth of the inference time (*e.g.*, 150 min. *vs*. 30 min. using SAM and Mask2Former on 2,573 data instances) while mostly achieving higher IoU performances across different methods and similar upper bound scores (*e.g.*, mIoU of 74.39 and 76.85 using SAM and Mask2Former, respectively).

| | Pretrained Models | RefCOCOg | | | RefCOCO | | | RefCOCO+ | | |
|---|---|---|---|---|---|---|---|---|---|---|
| | | val (U) | test (U) | val (G) | val | testA | testB | val | testA | testB |
| ***Zero-shot methods* (CLIP ViT-B/32)** | | | | | | | | | | |
| Global-Local Yu et al. (2023) | FreeSOLO | 31.11 | 30.96 | 30.69 | 24.88 | 23.61 | 24.66 | 26.16 | 24.90 | 25.83 |
| Global-Local Yu et al. (2023); Suo et al. (2023) | SAM | 27.57 | 27.87 | 27.80 | 22.43 | 24.66 | 21.27 | 26.35 | 30.80 | 22.65 |
| HybridGL† Liu & Li (2025) | SAM | 36.16 | 36.27 | 36.16 | 32.02 | 32.50 | 31.98 | 29.73 | 31.75 | 26.64 |
| TAS Suo et al. (2023) | SAM, BLIP-2 | 35.84 | 36.16 | 36.36 | 29.53 | 30.26 | 28.24 | 33.21 | 38.77 | 28.01 |
| CoPatch (Top-1) | Mask2Former | **41.43** | **41.98** | **41.37** | 37.17 | 44.63 | 31.53 | 35.15 | 43.14 | 29.97 |
| CoPatch (Top-3) | Mask2Former | 57.96 | 57.78 | 57.80 | 50.28 | 58.79 | 46.09 | 54.02 | 62.24 | 47.86 |
| ***Zero-shot methods* (CLIP ViT-B/16)** | | | | | | | | | | |
| Ref-Diff Ni et al. (2023) | SAM, SD | 38.62 | 37.50 | 38.82 | 35.16 | 37.44 | 34.50 | 35.56 | 38.66 | 31.40 |
| HybridGL Ni et al. (2023) | SAM | 42.47 | 42.97 | - | **41.81** | 44.52 | **38.50** | 35.74 | 41.43 | 30.90 |
| CoPatch (Top-1) | Mask2Former | **44.13** | **46.24** | **45.26** | 40.85 | 48.45 | 34.98 | 38.39 | 46.17 | 32.15 |
| CoPatch (Top-3) | Mask2Former | 61.09 | 63.61 | 63.03 | 52.92 | 60.79 | 47.27 | 55.46 | 63.32 | 48.06 |
| ***Zero-shot methods* (DFN ViT-H/14)** | | | | | | | | | | |
| CoPatch (Top-1) | Mask2Former | **46.21** | **46.03** | **46.09** | 41.24 | 48.88 | 36.15 | 38.10 | 45.11 | 33.14 |
| CoPatch (Top-3) | Mask2Former | 65.55 | 66.99 | 66.59 | 55.95 | 65.45 | 49.83 | 60.11 | 67.68 | 53.06 |

Table 7: **Comparison of oIoU performances across zero-shot RIS methods on RefCOCOg, RefCOCO, and RefCOCO+ datasets**. We attain the overall best performances for CLIP (ViT-B/32 and ViT-B/16) and DFN (ViT-H/14), outperforming the SoTA, TAS Suo et al. (2023), and HybridGL Liu & Li (2025). Note that we † stands for the reproduced results.

## C.3 oIoU Performance

Table 7 shows the oIoU performances of various zero-shot RIS methods, including ours, across RefCOCO datasets. Similar to the mIoU results, CoPatch achieves the best overall results for all the backbones. Notably, CoPatch (CLIP ViT-B/32) achieves oIoUs of 41.98, 44.63, and 43.14 in RefCOCOg, RefCOCO, and RefCOCO+ test sets, which are +5.82p, +14.37p, and +4.37p higher than the previous SoTA method, TAS (Suo et al., 2023).

## C.4 Qualitative Comparison

Additional qualitative results of comparing Global-Local (+CT) and CoPatch (top-1) are illustrated in Figures 19 and 20. While Figure 19 shows several samples that our approach accurately segments by predicting the top-1 candidate mask correctly, Figure 20 shows several challenging cases even using our approach. For example, the first row sample in Figure 20 is segmented on the 'the first half of the sandwich to the '*right*' instead of '*left*.' for top-1 and top-3 predictions of CoPatch. Despite this, the top candidate masks produced using our method are much diverse compared to those produced using the Global-Local method Yu et al. (2023).

## C.5 Context Token Effect

We also show qualitative results of the effects of our context tokens on the segmented results in Figure 21. We highlight the context tokens in red that are added as the local-level features of the hybrid text features described in Section 3.1. As noted, the model could better predict the primary object from the referring expression, given the context of colors. For instance, the fourth sample in Figure 21 guides a model to locate the bike nearest to the 'green' helmet. The last sample also corrects the model to focus on the 'black' property more than the woman wearing a dark navy jacket. We leave as future work to discover which of the context tokens within the referring expression could better guide the model for zero-shot RIS tasks.

## C.6 Spatial Subset Generation

The prompt for instructing Qwen2.5-14B-Instruct (Yang et al., 2024) to classify each referring expression into 'Spatial' or 'Non-Spatial' depending on the presence of a spatial cue is in Table 8. We provide important clarifications for the model to distinguish words that have multiple definitions (*e.g.*, right, in) and several examples for classification. The list of unique spatial cues ($n = 950$) found using LLM for all the RefCOCOg, RefCOCO, RefCOCO+ validation datasets is as follows[1]:

---

[1]Due to the space limit, we provide the first 173 cues only.

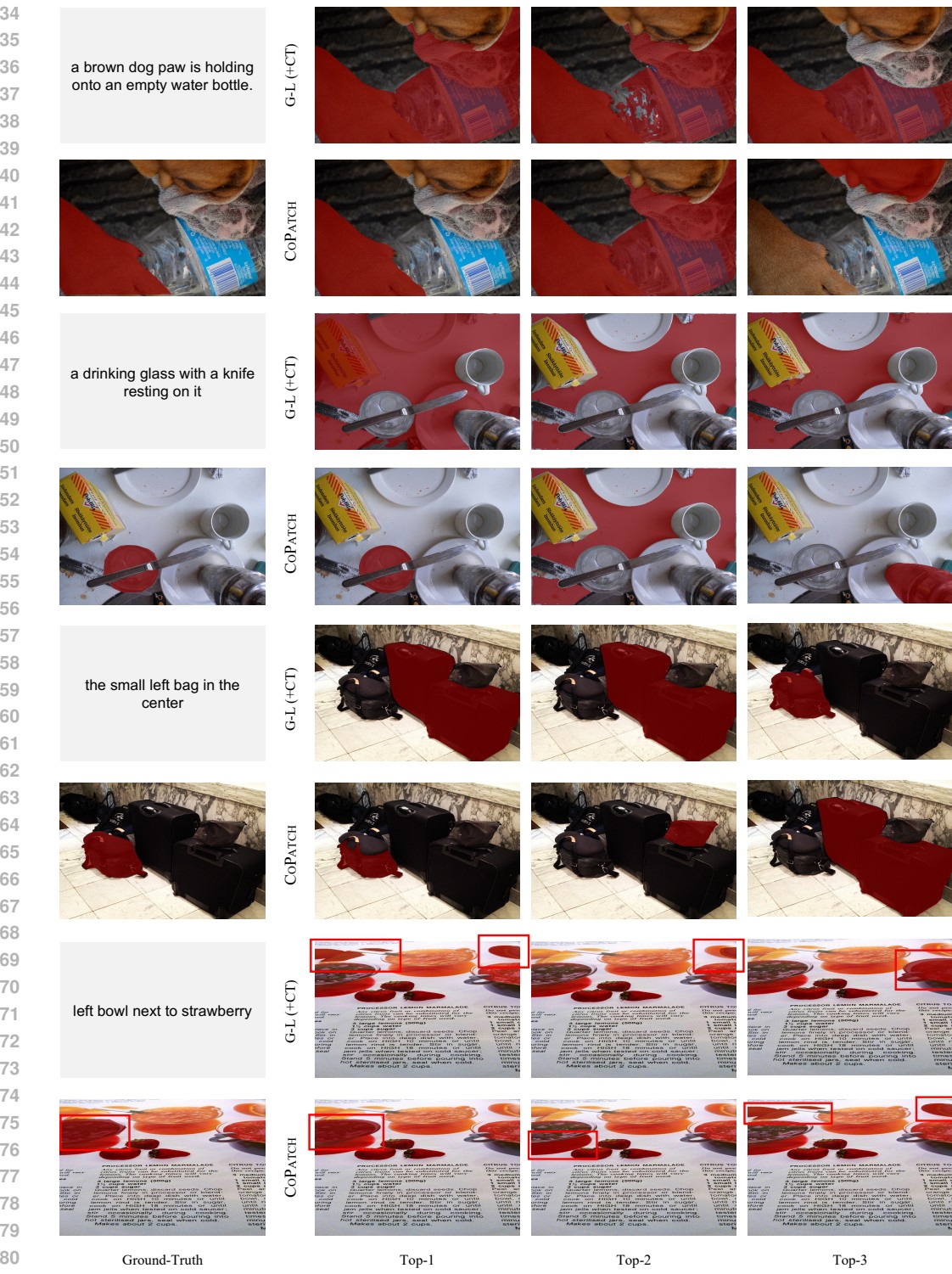

Figure 19: **Qualitative comparison of top candidate masks generated using original image-text similarity from Global-Local (w/ CT) and averaged CoMap similarity score using our CO-PATCH**. The top candidate masks generated using our method are much more diverse and accurate.

'12 o'clock', '2nd from', '2nd from bottom', '3rd from bottom', '3rd one from', 'None.', 'above', 'above left of', 'above right', 'above, left of', 'above, on, right', 'above, right', 'above, right of',

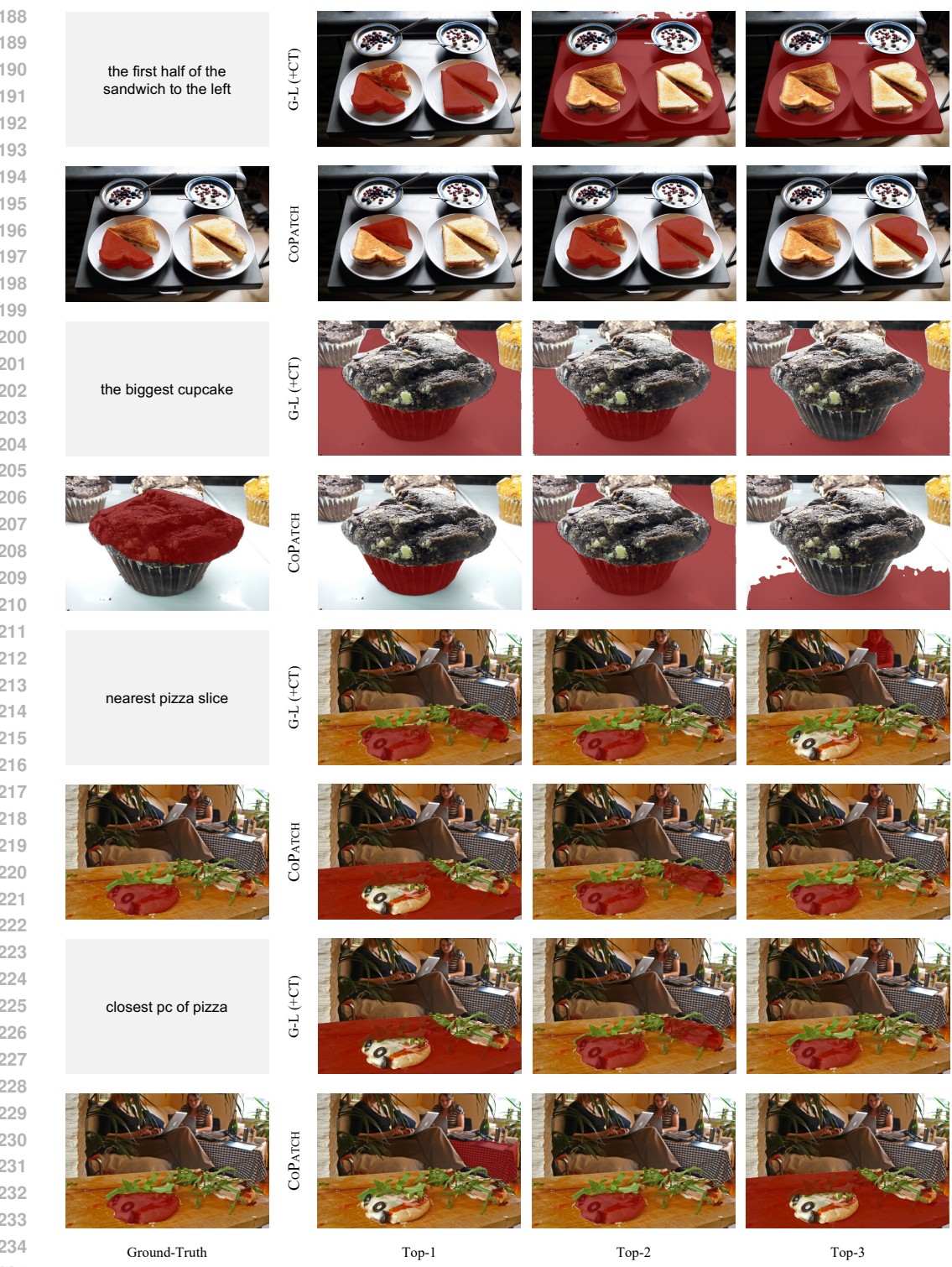

Figure 20: **Qualitative comparison of top candidate masks generated using original image-text similarity from Global-Local (w/ CT) and averaged `CoMap` similarity score using our COPATCH**. The top candidate masks generated using our method are much more diverse.

'across', 'adjacent to', 'after', 'against', 'against, on top of', 'ahead of', 'alongside', 'amongst', 'around', 'around it', 'at', 'at 4 o'clock', 'at 9 o clock', 'at the edge of', 'at the end of', 'at the end

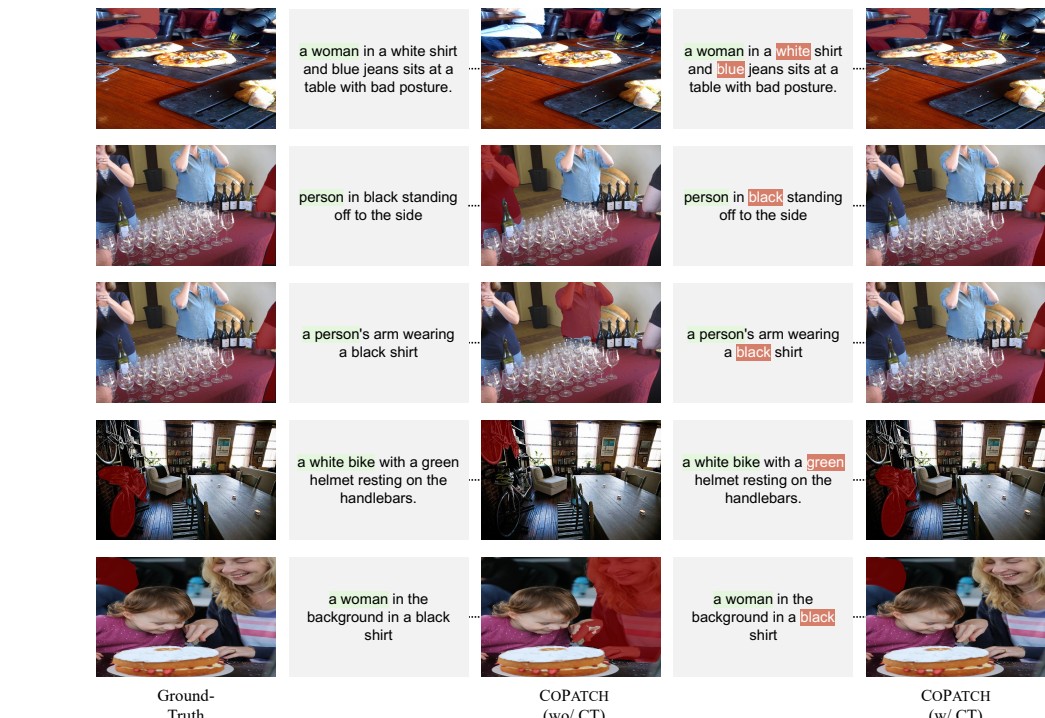

Figure 21: **Qualitative comparison between CoPatch without and with context tokens (CT).** We observe an improved mask candidate using hybrid text features highlighted with proposed context tokens in the local-level text features.

of, on the left', 'atop', 'attached to', 'back', 'back behind', 'back end of', 'back end of, on, right edge of', 'back end... front', 'back from the right', 'back left', 'back left of', 'back middle', 'back of', 'back of chair center top', 'back of the chair on the right', 'back of the pack', 'back of the room right side', 'back of, closest to', 'back of, far right', 'back of, flanking it', 'back of, in front of', 'back of, in the middle, behind', 'back of, left', 'back of, on, left', 'back on right', 'back on the right', 'back right', 'back right top', 'back row 4th from left', 'back row on left', 'back side', 'back side position', 'back top almost middle', 'back up high', 'back, in the front of', 'back, on', 'back, right', 'back, to the right', 'back... right', 'background', 'background center', 'background, far left', 'background, on', 'background, top left', 'backleft', 'backside', 'behidn', 'behind', 'behind and right of', 'behind middle', 'behind right of', 'behind right side', 'behind, front', 'behind, in, middle', 'behind, in, near', 'behind, left', 'behind, left of', 'behind, left shoulder of', 'behind, next to', 'behind, next to, on', 'behind, on the right', 'behind, on, left', 'behind, on, left side', 'behind, on, left, near', 'behind, right', 'behind, right side', 'behind, to the left of', 'below', 'below, above', 'below, in', 'below, left', 'below, on, left', 'beneath', 'beneath, on', 'beside', 'beside, closest to', 'beside, in front of', 'between', 'between, on', 'between, on the right', 'between, on, bottom left of', 'between, on, right', 'blocked by', 'blocking', 'bottom', 'bottom center', 'bottom from top', 'bottom front', 'bottom front left', 'bottom left', 'bottom left area', 'bottom left corner', 'bottom left corner of', 'bottom left of', 'bottom left portion of couch on left', 'bottom left side', 'bottom left, on', 'bottom leftcorner', 'bottom middle', 'bottom most', 'bottom most left', 'bottom of', 'bottom portion of it', 'bottom right', 'bottom right corner', 'bottom right corner of pic', 'bottom right front', 'bottom right next to', 'bottom right of', 'bottom right on', 'bottom right under', 'bottom rightcorner', 'bottom rightmost', 'bottom row', 'bottom row 2nd from left', 'bottom row second from left', 'bottom row second kid from left', 'bottom row, left', 'bottom to right', 'bottom, 2nd from top', 'bottom, first row, closest to', 'bottom, left', 'bottom, on, top, left', 'bottom-left', 'bottom-right', 'bottom-right, closest to', 'bottom/center', 'bottomright', 'by', 'by the wall', 'by the, on, left', 'by, with, to', 'center', 'center directly in front of', 'center front', 'center left', 'center leftish', 'center of', 'center of pic', 'center right', 'center row far right', 'center, behind', 'center/background', 'centermost', 'centernearest.'

---

**System and User Prompts for Spatial Subset Generation**

You are Qwen, created by Alibaba Cloud.
You are a helpful assistant.
Please carefully analyze user sentences to see if they contain spatial words or phrases.
Spatial expressions are any words or phrases indicating location, direction, orientation, or proximity.
These include (but are not limited to): 'on', 'in' (meaning 'inside'), 'under', 'front', 'behind', 'middle',
'center', 'left', 'right' (direction), 'closest to', 'near', 'beside', 'above', 'below', 'next to', etc.

**Important clarifications**:
- If 'right' means 'correct', it is NOT spatial.
- If 'in' means 'wearing' (e.g., 'in a costume'), it is NOT spatial.
- Action expressions like 'facing', 'looking up' is NOT spatial.
- 'closest to','near', or 'middle' should be treated as spatial. Respond with a specific format so it is easy to
parse.'

- - - - - - - - - - - - - - - - - - - - - - - - - - - - - - - - - - - - - - - - - - - - - - - - - - - - - - - -

Below are examples of how to classify:

Example 1:
Sentence: "The chair with the stuffed animal owl sitting in it."
Analysis: 'in' here is adverb.
Answer: IsSpatial: False, SpatialExpression: None

Example 2:
Sentence: "The apple is on the table."
Analysis: 'on' indicates spatial.
Answer: IsSpatial: True, SpatialExpression: on

Example 3:
Sentence: "left side monitor"
Analysis: 'left' indicates spatial.
Answer: IsSpatial: True, SpatialExpression: left

Example 4:
Sentence: "A small lamb lying closest to the adult."
Analysis: 'closest to' indicates a spatial relationship.
Answer: IsSpatial: True, SpatialExpression: closest to

Now, analyze this new sentence:
{sentence}

**Instructions**:
1) If the sentence contains any spatial relationship expression, output True and the exact expression (e.g.,
'on', 'in', 'closest to'). Important: actions or wearing attributes are NOT spatial expression. (e.g. a girl
facing the camera. -¿ 'facing' is NOT means direction or spatial relationship)
2) If no spatial expression is found, output False and None.
3) Return your answer in the format exactly: "IsSpatial: ¡True/False¿, SpatialExpression: ¡expression or
None¿"

Table 8: **Prompts used for generating spatial subsets of RefCOCO, RefCOCO+, and Ref-
COCOg validation sets**. We prompt Qwen2.5-14B-Instruct to divide the samples into Spatial and
Non-Spatial subsets.

C.7 HYPERPARAMETER TUNING

We show hyperparameter tuning results for COPATCH (CLIP ViT-B/32) and COPATCH (DFN ViT-
H/14) in Figures 22 and 23, respectively. Similar to COPATCH (CLIP ViT-B/16) results, the IoU
performances are not particularly sensitive to certain hyperparameter values. We select the exit layer
($l$) towards the last layer, yielding the best top-1 and top-3 performances. That means, for COPATCH
(CLIP ViT-B/32) and COPATCH (DFN ViT-H/14), we select the 10-*th* ($> 6$) and 22-*th* ($> 18$) exit
layer with the highest top-3 and top-1 performances. We select the initial threshold ($\delta$) based on
the top-3 performance for all three models. Lastly, since CoMap is not affected by the proportion

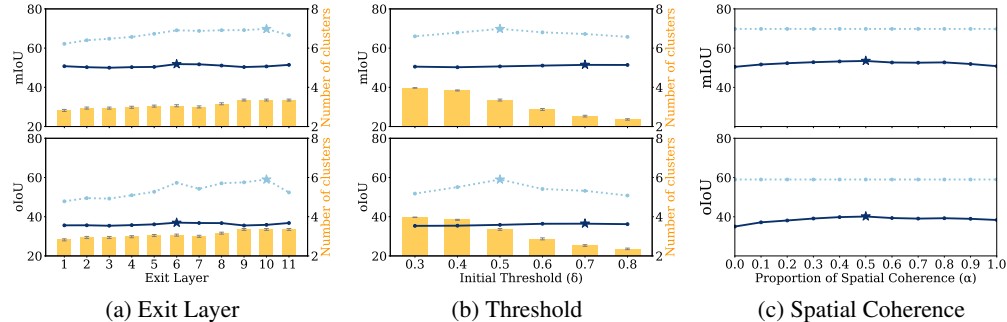

Figure 22: **Hyperparameter tuning results for CoPATCH (ViT-B/32) using 10% of the Ref-COCOg val set**. The **deep-blue** and **sky-blue** lines depict **top-1** and **top-3** performances across exit layers. The **golden** bars represent the number of clusters averaged across data instances for each exit layer. We set the exit layer, initial threshold, and spatial coherence proportion to be 10, 0.5, and 0.5.

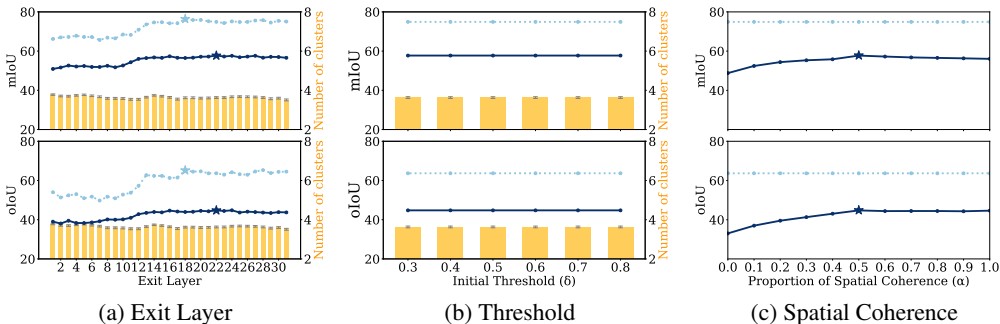

Figure 23: **Hyperparameter tuning results for CoPATCH (ViT-B/32) using 10% of the Ref-COCOg val set**. The **deep-blue** and **sky-blue** lines depict **top-1** and **top-3** performances across exit layers. The **golden** bars represent the number of clusters averaged across data instances for each exit layer. We set the exit layer and spatial coherence proportion to be 22 and 0.5. Since the performance does not change depending on the initial threshold, we set it to the default setting ($\delta = 0.5$).

of spatial coherence ($\alpha$), $\alpha$ is selected based on the top-1 performance. We set the exit layer, initial threshold, and spatial coherence proportion to be 8, 0.3, and 0.7 for CoPATCH (CLIP ViT-B/16) for all the performance results throughout the paper.

Note that the average number of clusters is not affected by the exit layer. However, it intuitively decreases as the initial threshold increases, only for CoPATCH (CLIP ViT-B/16) and CoPATCH (CLIP ViT-B/32). CoPATCH (DFN ViT-H/14) is possibly less influenced due to the increased patch number (16) (see visualizations in Figure 14).

### C.8 ALTERNATIVE TEXT FEATURE FUSION METHODS

We construct hybrid text features based on a weighted sum of global- and local-level text features, following previous works Yu et al. (2023); Liu & Li (2025). Here, we show results using an alternative fusion strategy of combining the features at an attention level. Specifically, instead of merging the text features at the final layer, we encode the sentence-level (global) and primary noun and context token-level (local) text features separately until the fusion layer and merge the attention outputs (*i.e.*, fused$_{attn}$ + $\gamma$global$_{attn}$ + $(1 - \gamma)$local$_{attn}$) before encoding them to construct the hybrid text features. Note that we use the fused features for the remaining layers after going through the fusion layer. As shown in Table 9 ($\gamma = 0.5$, we find that merging the text features as a weighted sum yields better performance in RefCOCOg.

| Method | Fusion Layer | RefCOCOg (oIoU / mIoU) | RefCOCO (oIoU / mIoU) | RefCOCO+ (oIoU / mIoU) |
|---|---|---|---|---|
| ATTN | 1 | 31.65 / 42.07 | 34.57 / 41.05 | 36.32 / 41.89 |
| ATTN | 2 | 33.33 / 43.73 | 36.11 / 43.96 | 36.69 / 42.77 |
| ATTN | 3 | 24.80 / 35.20 | 30.03 / 37.37 | 30.29 / 35.33 |
| ATTN | 4 | 31.36 / 43.89 | 34.29 / 41.39 | 34.01 / 39.71 |
| ATTN | 5 | 33.13 / 45.45 | 34.92 / 42.64 | 33.80 / 38.52 |
| ATTN | 6 | 35.47 / 49.12 | 36.24 / 44.01 | 35.82 / 41.65 |
| ATTN | 7 | 38.70 / **51.22** | 36.99 / 44.67 | 33.80 / 39.57 |
| ATTN | 8 | 38.65 / 51.05 | 35.62 / 45.15 | **36.09 / 44.65** |
| ATTN | 9 | 31.94 / 48.17 | **39.56 / 48.36** | 35.42 / 42.77 |
| ATTN | 10 | 35.37 / 50.52 | 35.35 / 44.05 | 35.56 / 43.32 |
| ATTN | 11 | 35.85 / 51.09 | 31.09 / 41.53 | 34.50 / 42.07 |
| WS | – | **39.07 / 51.22** | 36.46 / 45.35 | 35.63 / 43.73 |

Table 9: **Performance comparison between attention-based (ATTN) and weighted-sum-based (WS) fusion strategies across RefCOCOg, RefCOCO, and RefCOCO+ (10%) validation datasets**. We observe that the best performance is achieved using exit layers from the middle layers that are closer to the final layers for both fusion methods.

| Exit Layer | Entire RefCOCOg (val) oIoU / mIoU | 10% RefCOCOg (val) oIoU / mIoU | 10% RefCOCO (val) oIoU / mIoU | 10% RefCOCO (testB) oIoU / mIoU |
|---|---|---|---|---|
| 1 | 36.00 / 47.20 | 37.60 / 51.59 | 35.64 / 43.97 | 28.46 / 36.89 |
| 2 | 36.75 / 47.59 | 38.07 / 52.51 | 36.04 / 44.41 | 28.94 / 37.17 |
| 3 | 37.02 / 47.95 | 38.20 / 52.94 | 34.01 / 42.96 | 29.48 / 38.45 |
| 4 | 36.59 / 48.18 | 37.64 / 52.20 | 33.47 / 41.93 | 30.53 / 40.36 |
| 5 | 37.17 / 48.60 | 37.84 / 52.50 | 32.64 / 40.57 | 31.15 / 41.00 |
| 6 | 37.52 / 49.21 | 38.83 / 52.90 | 34.75 / 42.69 | **31.56 / 43.35** |
| 7 | 38.45 / 50.62 | **38.90** / 52.97 | 37.02 / 45.39 | 31.37 / 42.47 |
| 8 | **39.09** / 51.14 | 38.65 / 52.81 | 38.41 / **48.90** | 30.99 / 41.66 |
| 9 | 38.77 / 51.02 | 37.85 / 52.62 | **38.96** / 48.05 | 30.45 / 40.63 |
| 10 | 39.07 / **51.22** | 38.46 / **52.98** | 36.46 / 45.35 | 29.28 / 40.21 |
| 11 | 39.63 / 50.61 | 37.81 / 52.09 | 34.89 / 44.30 | 31.09 / 41.53 |

Table 10: **Layer-wise performance trends on different types of datasets**. We observe that using middle-to-late layers near the final layer generally provides the most generalizable spatial and semantic information.

| Model Type | RefCOCOg | | | RefCOCO | | | RefCOCO+ | | |
|---|---|---|---|---|---|---|---|---|---|
| | val (U) | test (U) | val (G) | val | testA | testB | val | testA | testB |
| *Zero-shot RIS methods* (BLIP) | | | | | | | | | |
| Global-Local (+CT) | 15.26 | 12.52 | 13.28 | 18.79 | 11.35 | 8.97 | 18.67 | 11.33 | 10.32 |
| CoPatch | 30.80 | 25.37 | 31.34 | 39.64 | 28.15 | 30.08 | 39.57 | 26.55 | 22.74 |
| *Zero-shot RIS methods* (SigLIP) | | | | | | | | | |
| Global-Local (+CT) | 36.31 | 36.68 | 37.00 | 39.24 | 43.99 | 36.17 | 35.23 | 40.44 | 30.23 |
| CoPatch | 39.12 | 39.85 | 41.02 | 43.10 | 48.72 | 39.44 | 38.97 | 45.90 | 34.21 |

Table 11: **Performance of non-CLIP models on RefCOCOg, RefCOCO, and RefCOCO+ datasets**. We show our method can be applicable to BLIP and SigLIP, achieving superior performance than the baseline - Globa-Local (+CT).

## C.9 EXIT LAYERS

Since the choice of exit layer is important, as it directly affects the overall quality of our spatial map, COPATCH, we show a performance change across different layers using the entire vs. 10% of the RefCOCOg (val) dataset, containing long, complex referring expressions, and 10% of the RefCOCO (val), and RefCOCO (testB) datasets, which contain various objects of the same category per image in relatively shorter referring expressions (Yang et al., 2022). Although the most optimal exit layer

slightly differs across the datasets, the selected exit layers per dataset/domain are mostly the middle layers near the final layers (6–11), as can be shown in Table 10.

### C.10 GENERALIZABILITY TO NON-CLIP-BASED VLMs

We use CLIP variants for a fair comparison with previous zero-shot RIS methods. However, to demonstrate the applicability of our method to other types of VLM, we show the results of applying our method to BLIP Li et al. (2022b) and SigLIP (Zhai et al., 2023) using Mask2Former. Whereas the final mask is selected based on the similarity between the original image and text embeddings for the Global-Local Yu et al. (2023) (+CT; Context Token) method, our method selects the best mask from top candidates (TC), reranked using our CoMap. As can be observed in Table 11, using our method consistently shows performance improvement compared to the Global-Local method.

## D DISCUSSION

### D.1 LIMITATIONS AND FUTURE WORK

We provide limitations of our framework and suggest future research directions for zero-shot RIS.

**Spatial Guidance.** The current spatial guider does not always select the correct candidate among different top candidate masks, leading to a gap in top-1 and top-$k$ IoU performances. Here, our goal is not to use any other additional models except for the pretrained CLIP backbone and mask generator. However, future work could explore an external spatial guide that can better select the final mask among the top candidate masks provided by our CoMap.

**Spatial Map.** The primary goal of this paper is to *provide spatial cues* for RIS tasks using the proposed spatial map, CoMap, using image embeddings extracted from intermediate layers. The current spatial map could serve as a baseline for *top candidate mask augmentation* to disperse the model's attention to look into multiple target objects located in different positions. We leave as a future work to extend the application of the presented spatial map into segmentation tasks that often require more refined segmentation, possibly with additional postprocessing methods (Shao et al., 2024; Bai et al., 2024).

### D.2 SOCIETAL IMPACT

The current study investigates zero-shot RIS, a task that requires a model to segment image regions based on free-form natural language descriptions–even for categories or concepts unseen during training. Without a need for additional training on explicit annotations, our method provides a flexible, adaptable approach applicable to many tasks related to spatial understanding. This zero-shot RIS system has the potential to significantly reduce reliance on costly human workers, paving the way for broader, scalable deployment An et al. (2023; 2025b). For instance, in healthcare, zero-shot RIS could aid radiologists to highlight specific regions of interest in medical scans (e.g., "the slightly darkened area near the lower part of the left lung") for detailed examination. Applying zero-shot RIS to recent advanced fields could enhance the effectiveness and accessibility of vision-language systems involving precise interpretation of free-form user input of referring expressions.

### D.3 ETHICAL AND REPRODUCIBILITY STATEMENT

Our work aims to leverage pretrained CLIP to improve general-purpose zero-shot RIS performance. The unintended bias may occur from the pretrained model itself; however, a comprehensive analysis and mitigation of such biases are beyond the scope of this work, which focuses on the core segmentation task. We provide data and code in the supplementary materials to ensure the reproducibility of our research.

