# OpenReview forum: "CoPatch: Zero-Shot Referring Image Segmentation by Leveraging Untapped Spatial Knowledge in CLIP"
_ICLR.cc/2026/Conference — Submitted to ICLR 2026_

### Official Review · Reviewer_ehKc · 2025-10-19

**Soundness:** 3
**Presentation:** 3
**Contribution:** 2
**Rating:** 4
**Confidence:** 4

**Summary:**

CoPatch is a training-free, zero-shot referring image segmentation (RIS) method. It aims to improve spatial grounding in CLIP by (1) constructing hybrid text features that augment the primary noun with context tokens (CT) (e.g., "woman smiling"), and (2) building a patch-level spatial map, CoMap, from intermediate CLIP visual layers. The CoMap is generated by computing patch-text cosine similarity, crucially applying a negation (sign-flip) to the patch embeddings before projection to correct for "opposite visualization" artifacts . This map is then clustered and used to re-rank candidate masks generated by an off-the-shelf model. The approach shows improved zero-shot mIoU over baselines on RefCOCO/+/g and PhraseCut. Its Top-3 performance is exceptionally strong (e.g., +16.28 mIoU over Top-1 on RefCOCOg val), suggesting the CoMap is highly effective at identifying good candidates, even if the final Top-1 selection is imperfect.

**Strengths:**

1. **Simple and well-motivated method**: Based on the experimental results, tapping intermediate layers for spatial knowledge and using a simple negation to fix inverted saliency maps are highly effective.

2. **Strong performance**: Significant boost from Top-1 to Top-3 results (Table 1) validates the core claim - CoMap is a superior guide for scoring and ranking mask proposals compared to standard global-local similarity.

3. **Good component analysis**: e.g. the ablations in Table 3(b) effectively demonstrate that each component—Context Tokens (CT), Top Candidate (TC) re-ranking, and Spatial Coherence (SC)—provides a measurable benefit.

**Weaknesses:**

1. **Concerns on Architectural Generalizability:** The method's generalizability is a key concern, as its effectiveness appears tightly coupled to the specific architecture and training objective of ViT-based CLIP models. It remains unclear if the CoMap framework can be extended to convolutional backbones, such as the convnext_large_d_320 variant of OpenCLIP. Furthermore, the "negation trick," which seems tailored to correct an artifact of CLIP's contrastive loss, may not be relevant or effective for models trained with different objectives, like the sigmoid-based loss used in ViT-SO400M-14-SigLIP-384. I would like to see experiments with these different variants.

2. **Missing quantitative ablation:**
    1. *Negation on/off*—The negation step is central to making CoMap work (Section B.2, Figure 11). However, there is no mIoU/oIoU ablation showing performance with versus without this negation. This is a critical piece of evidence needed to validate the component's true impact.
    2. *No “no-clustering” baseline*—How does the method perform if masks are scored directly with the raw CoMap (no connected-component clustering)?

3. **Confounded main comparisons:** The evaluation of CoPatch against key baselines like HybridGL and Global-Local is unfair. CoPatch is benchmarked using mask proposals from Mask2Former, while the baselines rely on proposals from SAM or FreeSOLO. Since the choice of mask generator significantly influences performance, this discrepancy prevents a fair assessment. While the appendix (Section C.2) attempts to address this issue, the analysis is undermined by the use of different evaluation metrics. Specifically, the results in Table 1 are reported in mIoU, whereas Table 6 uses oIoU, rendering the tables incomparable. To ensure a fair and transparent evaluation, the data from Tables 1, 6, and 7 should be reorganized into a unified analysis that uses a consistent metric across all methods. This direct comparison is crucial for understanding the true performance of CoPatch and should be presented in the main body of the paper—either in the primary results table or as a dedicated ablation study—rather than being relegated to the appendix.

4. **Lack of Discussion of Relevant Works:** The paper omits discussion and comparison with recent, highly relevant works—a significant oversight for a SOTA-focused paper. These works must be addressed in the related work section and included in the Table 2 comparisons.
    1. *Feature Design for Bridging SAM and CLIP (Ito et al., WACV 2025)*—A trained CLIP↔SAM bridge that mirrors CoPatch’s proposal + CLIP scoring pipeline.
    2. *Mask Grounding for Referring Image Segmentation (Chng et al., CVPR 2024)*—Supervised method that aligns word tokens to pixels, similar to same fine-grained text-to-pixel goal as CoPatch.

5. **Minor nits**:
    1.  Figure 13 Caption: "CLIP ViT-B/316" is a typo for "ViT-B/16".
    2.  Consistency: The method name should be unified (e.g., CoPatch without dash in the title vs. Co-Patch in Figure 5 and Figure 7).

**Questions:**

See weaknesses

---

> ### Author Response · Authors · 2025-11-19
>
> Thank you for your insightful comments and questions and for acknowledging our method, performance, and analysis. Please read our point-by-point responses, and let us know if there are any remaining issues you would like us to clarify further.
>
> **W1) Concerns on Architectural Generalizability**
>
> We would like to emphasize that our method is intentionally built on ViT-based CLIP models, as multiple studies [1–3] consistently demonstrate that ViT-based CLIPs achieve comparable or even superior zero-shot RIS performance relative to their CNN-based counterparts. We leave as future work to explore untapped spatial cues in CNN-based CLIP models. We will clearly state in the manuscript that the current method is evaluated using ViT-based CLIP models.
>
> We show the results using our method applied to SigLIP, as shown in the Table below. Because SigLIP does not have an embedding corresponding to [cls], unlike CLIP, it does not need the “negation trick” since the “inversion artifact” (line 228) is not present. Other than the negative computation, we apply the same method as described in the paper.
>
> |         | RefCOCOg (val-u) | RefCOCOg (test-u) | RefCOCO (val) | RefCOCO (testA) | RefCOCO (testB) | RefCOCO+ (val) | RefCOCO+ (testA) | RefCOCO+ (testB) |
> |---------|------------------|-------------------|---------------|-----------------|-----------------|----------------|------------------|------------------|
> | SIGLIP  | 36.31            | 36.68             | 39.24         | 43.99           | 36.17           | 35.23          | 40.44            | 30.23            |
> | COPATCH | 39.12            | 39.85             | 43.10         | 48.72           | 39.44           | 38.97          | 45.90            | 34.21            |
>
> Supporting our original claim, our method consistently yields improvements even when applied to SigLIP, a model trained with objectives that differ from CLIP. This further demonstrates that our approach is not tied to a specific architecture or training paradigm. Additionally, as noted in the main manuscript, we have already included results on DFN and BLIP, showing generalizability and robustness of our method across VLMs.
> Finally, we would like to re-emphasize that the core novelty of our work lies in **uncovering and leveraging CLIP’s previously untapped spatial knowledge for zero-shot RIS,** which is the central contribution highlighted in our paper’s title. We hope these clarifications help demonstrate the significance of our contribution.
>
> **W2) Missing quantitative ablation: Negation on/off**
>
> Yes, the negation step is central in creating the CoMap. Since creating the similarity map without negation would result in a non-intuitive similarity map (high scores for low patch-text similarity values and vice versa), the ablation results (mIoU/oIoU) are much lower than those of our final method, as shown in the Table below. Similarly, we find that using no clustering strategy also results in a performance drop compared to ours. We appreciate this comment to improve the validity of our method. We will include this experimental result in the revised version of the manuscript.
>
> |         |         |           |            |   RefCOCOg  |   RefCOCOg  |   RefCOCOg  |   RefCOCO   |   RefCOCO   |   RefCOCO   |   RefCOCO+  |   RefCOCO+  |   RefCOCO+  |
> |:-------:|:-------:|:---------:|:----------:|:-----------:|:-----------:|:-----------:|:-----------:|:-----------:|:-----------:|:-----------:|:-----------:|:-----------:|
> |         |         | Negation? | Clustering? | val (U)     | test (U)    | val (G)     | val         | testA       | testB       | val         | testA       | testB       |
> | COPATCH | (Top-1) | ✕         | ✓          | 49.53/40.53 | 50.67/42.34 | 49.41/41.71 | 40.15/33.92 | 44.52/40.06 | 35.56/31.69 | 41.99/34.53 | 46.38/40.12 | 35.36/31.01 |
> | COPATCH | (Top-3) | ✕         | ✓          | 54.20/43.07 | 55.50/45.30 | 54.34/44.31 | 43.74/34.63 | 47.94/40.72 | 40.01/31.98 | 46.70/36.69 | 50.09/42.30 | 41.55/32.94 |
> | COPATCH | (Top-1) | ✓         | ✕          | 50.00/39.86 | 51.16/42.48 | 49.91/41.13 | 41.52/34.41 | 47.45/41.12 | 35.23/30.43 | 42.45/34.39 | 47.29/39.63 | 34.77/29.57 |
> | COPATCH | (Top-3) | ✓         | ✕          | 57.68/44.96 | 58.51/47.22 | 57.95/46.01 | 49.35/37.92 | 55.80/47.32 | 44.10/32.83 | 52.52/40.53 | 57.34/48.97 | 45.99/34.00 |
> | COPATCH | (Top-1) | ✓         | ✓          | 54.42/44.13 | 55.62/46.24 | 54.71/45.26 | 50.05/40.85 | 55.38/48.45 | 43.01/34.98 | 46.83/38.39 | 54.08/46.17 | 37.67/32.15 |
> | COPATCH | (Top-3) | ✓         | ✓          | 71.21/61.09 | 72.29/63.61 | 72.11/63.03 | 65.38/52.92 | 67.77/60.79 | 62.45/47.27 | 67.41/55.46 | 69.86/63.32 | 63.83/48.06 |

---

> ### Author Response · Authors · 2025-11-19
>
> **W3) Confounded main comparisons**
>
> Thank you for your critical comment. As noted in lines 322-323, we provide results using the same pre-trained masks (SAM) as the SoTA methods in Appendix C.2. We clarify that both Tables 1 and 6 are reported in mIoU.
>
> The reason we have included the results using Mask2Former since there was no significant difference in performance using either Mask2Former or SAM, but only with a significant efficiency improvement using Mask2Former over SAM (x5 faster, as noted in lines 1044-1046).
>
> However, we fully agree with the reviewer’s suggestion that the comparison should be presented more clearly by using identical mask proposals in Table 2. In response to this, we will update the manuscript by reorganizing the tables and adding consistent metric comparisons directly in the main text.
>
> **W4) Lack of Discussion of Relevant Works**
>
> Thank you for providing relevant works. Since our work is targeting *zero-shot* RIS methods, we exclude training-based SoTA supervised methods in the main table. However, we will include these works in the first subsection of the Related Works.
>
> **W5) Minor nits**
>
> Sorry, and thank you for catching these typos. We will revise these in the revised version of the manuscript.
>
> *References*
>
> [1] Zero-shot referring image segmentation with global-local context features (CVPR, 2023).
>
> [2] Text Augmented Spatial-aware Zero-shot Referring Image Segmentation (Findings of EMNLP, 2023).
>
> [3] Hybrid global-local representation with augmented spatial guidance for zero-shot referring image segmentation (CVPR, 2025).
>
> **Thank you once again for carefully reviewing our work. We sincerely hope that our responses have resolved the concerns you raised, and we would greatly appreciate it if you could revise the score to reflect these clarifications. If there are any remaining issues or further details you would like us to address, please let us know and we will promptly provide additional explanation.**

---

### Official Review · Reviewer_veLK · 2025-10-30

**Soundness:** 2
**Presentation:** 2
**Contribution:** 1
**Rating:** 2
**Confidence:** 5

**Summary:**

This paper presents a method of leveraging spatial knowledge in the pretrained model to improve zero-shot referring image segmentation. In particular, the proposed CoPatch constructs hybrid text features by considering context tokens that carry spatial cues. In addition, CoPatch extracts patch-level image features using a novel path found from intermediate layers. Then, these enhanced features are employed to calculate image-text similarity map.

**Strengths:**

(1) The motivation is well presented of leveraging spatial knowledge in the pretrained model to improve zero-shot referring image segmentation.

(2) The illustrations is mostly clear and intuitive of the context tokens, the patch-level spatial map, the fusion and the reranking of top candidate masks.

**Weaknesses:**

(1) The contribution is limited. In particular, the task setting is not clearly presented where this zero-shot setting is not well distinguished with the supervised setting since a pretrained model is employed. In addition, the experimental comparison seems to be unfair and problematic.

(2) On Page 6, Line 292-293, the comparison is unfair since the proposed CoPatch uses Mask2Former as pretrained model while all the SOTA methods does not.

(3) In Table 1, Mask2Former Upper Bound performs much better than Mask2Former +CoPatch, it is wired and there has no explanation.

(4) On Page 7, Line 347-350, the oracle performance is not well explained which makes the comparison and discussion very confusing. In addition, the results are not well organized to show a clear comparison.

(5) On Page 7, Line 377, still on the oracle performance issue, the low performance is not mainly due to pretrained models. It seems to have nothing to do with the main experimental results.

**Questions:**

No.

---

> ### Author Response · Authors · 2025-11-18
>
> Thank you for your valuable time in reviewing our paper and recognizing our motivations to be well presented and our illustrations to be intuitive. Please read our point-by-point responses, and let us know if there are any remaining issues you would like us to clarify further.
>
> **W1) The contribution is limited. In particular, the task setting is not clearly presented where this zero-shot setting is not well distinguished with the supervised setting since a pretrained model is employed. In addition, the experimental comparison seems to be unfair and problematic.**
>
> We respectively disagree with the comment on our task setting not being clearly presented. As stated in the Introduction, zero-shot RIS refers to segmenting the referred object without any task-specific training. This setting follows the standard protocol used in recent works [1–4]. Using pretrained CLIP backbones and pretrained segmentation models does not violate the zero-shot setting, as these models are not trained for referring segmentation. They are only used to transfer general visual–language and segmentation capabilities, which is the common and established practice in zero-shot RIS. Regarding the concern about unfair experimental comparison, please refer to our response in W2 (below).
>
> **W2) On Page 6, Line 292-293, the comparison is unfair since the proposed CoPatch uses Mask2Former as pretrained model while all the SOTA methods does not.**
>
> As noted in lines 322-323, we have our results using the same pre-trained masks (SAM) as the SoTA methods in Appendix C.2. The reason we have included the results of our method using Mask2Former since there was no significant difference in performance using either Mask2Former or SAM, but only with a significant efficiency improvement using Mask2Former over SAM (x5 faster, as noted in lines 1044-1046). We acknowledge that it would have been better to include the results using the same masks in the main table for better clarity. We will incorporate it in the revised version of the manuscript.
>
> **W3, W4 & W5) In Table 1, Mask2Former Upper Bound performs much better than Mask2Former +CoPatch, it is wired and there has no explanation. On Page 7, Line 347-350, the oracle performance is not well explained which makes the comparison and discussion very confusing. In addition, the results are not well organized to show a clear comparison. On Page 7, Line 377, still on the oracle performance issue, the low performance is not mainly due to pretrained models. It seems to have nothing to do with the main experimental results.**
>
> As noted in the footnote on page 7, the upper bound (oracle) performance is computed as the IoU between the best mask generated by the pretrained segmentation model and the ground-truth mask. By definition, this upper bound is expected to be higher than our zero-shot results, since it uses the ground-truth mask to choose the optimal segmentation. We include this oracle score—consistent with the key prior work [1]—to show that the bottleneck of zero-shot RIS does not primarily come from the pretrained segmentation model. Rather, the main challenge lies in how effectively a VLM (e.g., CLIP) can select the correct mask among multiple proposals. The oracle serves to indicate how much performance could be achieved if this mask-selection step were perfect.
>
>
> *References*
>
> [1] Zero-shot referring image segmentation with global-local context features (CVPR, 2023).
>
> [2] Text augmented spatial aware zero-shot referring image segmentation (Findings of EMNLP, 2023).
>
> [3] Iterprime: Zero-shot referring image segmentation with iterative grad-cam refinement and primary word emphasis (AAAI, 2025).
>
> [4] Hybrid global-local representation with augmented spatial guidance for zero-shot referring image segmentation (CVPR, 2025).
>
> **We would be sincerely grateful if you could kindly reconsider the score, as we believe our clarifications directly address the concerns raised. We would also like to gladly respond to any other points that you want us to further clarify.**

---

### Official Review · Reviewer_MWds · 2025-11-01

**Soundness:** 3
**Presentation:** 3
**Contribution:** 2
**Rating:** 6
**Confidence:** 3

**Summary:**

The paper introduces COPATCH, a training-free framework for zero-shot referring image segmentation (RIS) that enhances spatial grounding by (i) augmenting CLIP’s local text features with context tokens and (ii) extracting patch-level image features from an intermediate visual layer to build a context-aware spatial map (CoMap). The method clusters high-similarity patches and uses the clusters to re-rank candidate masks from a pre-trained segmenter, then applies a spatial coherence filter when explicit positional cues are present. The authors argue that CLIP is not intrinsically “spatially blind”; rather, spatial information exists in intermediate representations and can be surfaced without the need for gradient or attention maps. Experimentally, COPATCH achieves strong zero-shot mIoU across RefCOCO/+/g and PhraseCut, often surpassing prior zero-shot and even weakly/pseudo-supervised baselines; top-1 improvements range roughly +2–7 mIoU, with much larger margins when reporting top-3 oracle metrics. The approach is hyperparameter-light (exit layer, threshold, and spatial coherence weight) and is claimed to require only a single forward pass for the map. Qualitative examples demonstrate better localization than Grad-CAM-based pipelines, and ablation analysis attributes the gains to both context-token text features and CoMap-guided candidate selection.

**Strengths:**

- Leveraging intermediate CLIP patch embeddings, along with lightweight clustering, to recover spatial cues is elegant and avoids fine-tuning or the use of expensive gradient maps. The paper provides evidence (in the form of layerwise similarity trends and latency comparisons) that this approach is both practical and efficient.
- Results on RefCOCO/+/g and PhraseCut demonstrate state-of-the-art zero-shot performance across multiple CLIP/DFN backbones. Ablations isolate the contributions of context tokens (CT), top-candidate (TC) re-ranking via CoMap, and spatial coherence (SC).
- Visualizations demonstrate a sharper focus on the referred object than Grad-CAM-based IteRPrimE; the spatial/non-spatial subset analysis strengthens the claim that both spatial grounding and semantics improve.

**Weaknesses:**

- Many baselines utilize SAM or FreeSOLO proposals, whereas COPATCH employs Mask2Former and also reports a Mask2Former upper bound. Without re-running baselines under the same proposal pool, some gains may reflect differences in candidate quality rather than scoring alone. A stronger fairness control would standardize the mask generator across methods.
- The paper frequently highlights considerable “Top-3” mIoU improvements, which reflect whether the correct mask appears among the top-k, not whether the method selects it. Unless coupled with an interaction or an automatic picking rule, these numbers are diagnostic rather than directly actionable.
- The negation of patch embeddings to fix an “inversion artifact,” the clustering threshold δ, and selection logic (including negative text features + SC) are plausible but somewhat ad-hoc; theoretical motivation or broader sensitivity analysis (beyond the 10% val tuning on RefCOCOg) would improve confidence in robustness.
- The extraction of “primary noun + context tokens” hinges on chunking/parsing rules and English expressions; the paper does not show multilingual or out-of-grammar robustness, and it is unclear how failures in chunking affect CoMap and mask scoring. (Appendix references and tool citations suggest rule-based tokenization choices, but details are light in the main text.)

**Questions:**

- Can the authors re-run the strongest prior zero-shot baselines using the same Mask2Former proposal set to isolate scoring improvements from proposal quality? If not, can they at least show COPATCH scoring on SAM proposals to bridge the comparison?
- How sensitive is the context-token extraction to parsing errors or non-English phrasing (e.g., multilingual RefCOCO variants)? Are there any failure analyses that show when CT helps versus when it hurts?

---

> ### Author Response · Authors · 2025-11-21
>
> Thank you for your thoughtful feedback and for recognizing the practicability and efficiency, as well as the impressive performance improvement using our method. Please read our point-by-point responses, and let us know if there are any remaining issues you would like us to clarify further.
>
> **W1 & Q1) Many baselines utilize SAM or FreeSOLO proposals, whereas COPATCH employs Mask2Former and also reports a Mask2Former upper bound. Without re-running baselines under the same proposal pool, some gains may reflect differences in candidate quality rather than scoring alone. A stronger fairness control would standardize the mask generator across methods. Can the authors re-run the strongest prior zero-shot baselines using the same Mask2Former proposal set to isolate scoring improvements from proposal quality? If not, can they at least show COPATCH scoring on SAM proposals to bridge the comparison?**
>
> Thank you for your critical comment. As noted in lines 322-323, we provide results using the same pre-trained masks (SAM) as the SoTA methods in Appendix C.2. We clarify that both Tables 1 and 6 are reported in mIoU.
>
> The reason we have included the results using Mask2Former since there was no significant difference in performance using either Mask2Former or SAM, but only with a significant efficiency improvement using Mask2Former over SAM (x5 faster, as noted in lines 1044-1046).
>
> However, we fully agree with the reviewer’s suggestion that the comparison should be presented more clearly by using identical mask proposals in Table 2. In response to this, we will update the manuscript by reorganizing the tables and adding consistent metric comparisons directly in the main text.
>
> **W2) The paper frequently highlights considerable “Top-3” mIoU improvements, which reflect whether the correct mask appears among the top-k, not whether the method selects it. Unless coupled with an interaction or an automatic picking rule, these numbers are diagnostic rather than directly actionable.**
>
> Yes, we acknowledge that the Top-3 performance is not directly actionable since it implies whether the correct mask is included among the top-3 candidates. We reported these top-k results to diagnose and emphasize the effectiveness of our Top Candidate (TC) component.
>
> Our motivation comes from observing that previous methods–based on the vanilla cosine similarity between the original image and text features–often generate highly overlapping mask candidates. By applying TC on our novel spatial map, which leverages previously untapped spatial knowledge in CLIP, our method reduces this overlap (see Fig. 8). We will put more emphasis on the Top-1 performance, which more accurately reflects the practical impact of our method.

---

> > ### Author Response · Authors · 2025-11-21
> >
> > **W3) The negation of patch embeddings to fix an “inversion artifact,” the clustering threshold δ, and selection logic (including negative text in features + SC) are plausible but somewhat ad-hoc; theoretical motivation or broader sensitivity analysis (beyond the 10% val tuning on RefCOCOg) would improve confidence in robustness.**
> >
> > We agree that the negation of patch embeddings, the clustering threshold, and the selection logic may appear ad hoc at first impression. As you understood correctly, we have performed the sensitivity analysis to empirically prove that our method is not sensitive to the *key hyperparameters* of CoPatch’s components. However, each of the CoPatch’s components is not ad hoc and directly helps to resolve the observed failure modes of CLIP. We explain the design choices for each component in more detail below:
> >
> > - **The negation**: Same as the finding from CLIP Surgery [1], the similarity between raw image patch embeddings and text embeddings exhibits “opposite visualizations,” where their embedding direction is inverted relative to the text concept. Instead of recomputing the attentions like [1], we simply find a way to negate the input vector (built on the cosine similarity property–please refer to Section B.2 in the Appendix) to create a “non-opposite visualization,” allowing the generated CoMap to be aligned with the intended semantic direction. Unlike previous zero-shot RIS works [2-3], which use gradients (GradCAM) or attention-level features for creating the map, we find a very simple, yet effective operation that directly addresses the core problem of CLIP when creating a similarity map with original embeddings rather than serving as an arbitrary heuristic.
> >
> > - **The clustering threshold**: We note that this is a hyperparameter (not the component) used for creating an intuitive, spatial map (CoMap). Our algorithm aims to return the interpolated, patch-level spatial map (Algorithm 1) based on the patch-level similarity (i.e., similarity between embeddings corresponding to different patches) from the initial similarity map. Hence, we need to decide whether two patches are similar or not, depending on the clustering threshold to assign each patch to a certain cluster. The threshold thus serves as a necessary criterion for converting continuous similarity values into discrete, spatially coherent regions, which is a standard practice in similarity-based clustering. Since we observe consistent behavior across a broad δ range, this suggests that the clustering process is not ad hoc, but a natural procedure that reinforces the inherent grouping structure already encoded in the patch-level similarities.
> >
> > - **The selection logic**: Calculating the similarity between positive and negative features was necessary to decide the positional relationship for the same objects located at different positions. We directly adopt the method from [3] and use our enhanced features to enhance the final mask selection, reflected in top-1 mIoU/oIoU performance.
> >
> > We will clarify the motivation behind each of these components/hyperparameters in the revised manuscript. Thank you for this feedback.

---

> ### Author Response · Authors · 2025-11-21
>
> **W4 & Q2) The extraction of “primary noun + context tokens” hinges on chunking/parsing rules and English expressions; the paper does not show multilingual or out-of-grammar robustness, and it is unclear how failures in chunking affect CoMap and mask scoring. (Appendix references and tool citations suggest rule-based tokenization choices, but details are light in the main text.) How sensitive is the context-token extraction to parsing errors or non-English phrasing (e.g., multilingual RefCOCO variants)? Are there any failure analyses that show when CT helps versus when it hurts?**
>
> Thank you for your comment regarding the robustness of our “primary noun + context token” (CT) extraction to multilingual or out-of-grammar inputs. We acknowledge that our main evaluation datasets, RefCOCO variants, and PhraseCut, are English-only, so we cannot directly assess multilingual robustness with our current CLIP-based pipeline. However, we understand your concern about how failures in chunking might affect CoMap and mask scoring. Hence, we further investigate failure scenarios to understand the robustness of CT extraction to parsing errors. We perform analysis on our strongest model, DFN (dataset: RefCOCOg val), to ensure that any observed failures reflect the robustness of the CT extraction itself rather than limitations of the underlying model.
>
> | **Grammatical Category**               | **Count** | **Mean IoU** |
> |----------------------------------------|-----------|--------------|
> | Full sentence with no punctuation              | 26        | 0.658        |
> | Fragmented sentence (noun phrase only) with punctuation            | 60       | 0.578        |
> | Full sentence with punctuation | 31        | 0.559        |
> | Fragmented sentence without punctuation | 68        | 0.513        |
>
> As can be observed in the table above, prompt inputs categorized as "Full Sentence with no punctuation" achieved the highest performance (0.658 Mean IoU), significantly outperforming "Fragmented sentence with punctuation" inputs (0.578 Mean IoU). This indicates that the presence of verbs and relational context, even in the absence of formal punctuation, provides the critical disambiguation cues that our model needs.
>
> Similarly, it is also interesting to notice that the performance corresponding to "Full Sentence with no punctuation" achieved significantly higher accuracy than that of the "Full Sentence (FS) with punctuation". This might be because the lengthy prompts often have punctuations (10.9 words in FS w/ punctuation > 8.9 words in FS wo/ punctuation), and overly specific, long phrases make it difficult for the model to understand.
>
> Overall, while CT extraction may underperform on fragmented sentences and sentences without punctuation, this limitation reflects the inherent ambiguity of such inputs rather than shortcomings of our method. We will add the analysis to the revised paper to avoid overly stating the robustness of the CT method.
>
>
> *References*
>
> [1] A Closer Look at the Explainability of Contrastive Language-Image Pre-training (Pattern Recognition, 2025).
>
> [2] Iterprime: Zero-shot referring image segmentation with iterative grad-cam refinement and primary word emphasis (AAAI, 2025).
>
> [3] Hybrid global-local representation with augmented spatial guidance for zero-shot referring image segmentation (CVPR, 2025).
>
> **Thank you again for your thorough review of our work. We hope our responses have addressed your concerns. Should any questions remain or if further details are needed, we would be happy to provide additional explanations.**

---

### Author Response · Authors · 2025-11-28
**General Response**

We sincerely thank all reviewers for their time and insightful feedback. With less than a week remaining in the review period, we would like to briefly follow up with this general response, clarifying our main contributions and summarizing how we have addressed common concerns.

Please check the updated PDF file, where we have highlighted the changes in **blue**. Also, we have added Table 4 (negation & clustering ablation) and Figure 9 (sensitivity of CT) to incorporate the reviewers’ feedback.

> ## **Main Contributions**

- **Novel zero-shot RIS framework**: CoPatch, a zero-shot referring image segmentation (RIS) method leveraging untapped spatial knowledge in CLIP.


- **Text feature enhancement**: Incorporates context tokens alongside the primary noun to better capture spatial cues in referring expressions.


- **Image feature extraction**: Finds a novel, computationally efficient pathway to extract patch-level image features used for building an intuitive, clustered image–text similarity map, CoMap.


- **SoTA performance**: Improves zero-shot mIoU across RefCOCO, RefCOCO+, RefCOCOg, and PhraseCut by +2–7 mIoU compared to the recent SoTA method, HybridGL, without additional training.


- **Ablation studies**: Studies the effect of the exit layers, thresholds, spatial coherence hyperparameter, along with negation and clustering (added during this rebuttal).


- **Generalization**: Experiments other than vanilla CLIP (DFN, BLIP, and SigLIP) demonstrate robust performance across various VLM architectures trained with different learning objectives.


> ## **Revisions and Clarifications**

- **Fair comparisons**: We put the results using the same pre-trained masks (SAM) as prior works on the main table (instead of the Appendix), alongside Mask2Former, demonstrating similar performance with substantial efficiency gains.

- **Design choices and hyperparameters**: We clarify the motivation and robustness of negation, clustering threshold, and selection logic.

- **Context-token (CT) robustness**: We analyze the effect of different grammatical inputs on performance to test the sensitivity of CT.

- **Metrics and oracle clarification**: Top-1 performance captures practical impact, while Top-3 demonstrates Top Candidate effectiveness. The explanation of the oracle performance is rewritten.

- **Minor issues**: Supervised RIS methods are added to the Related Works, and we have changed Co-Patch to CoPatch in all the figures.

> ## **Conclusion**

CoPatch is a simple, effective framework for zero-shot RIS that uncovers and leverages spatial knowledge in “spatially blind” encoder-based VLM, providing practical improvements without additional training. We hope these clarifications address the reviewers’ concerns and would greatly appreciate reconsideration of our scores.

---

### Meta-Review · Area_Chair_ggSL · 2026-01-09

**Summary:**

This paper proposes a training-free framework for zero-shot referring image segmentation that exploits intermediate CLIP representations and patch-level similarity maps to improve spatial grounding. The reviewers highlight the intuitive idea and the extensive empirical validation. However, they have also raised concerns regarding limited novelty (veLK), ad-hoc design components (MWds, ehKc) fairness issues arising from the use different methods for mask proposals (all), questions about architectural generalizability (ehKc), and the practical role of Top-3 metric (MWds).
The authors provided detailed responses to the reviews, and many concerns are well addressed. For example, the issue of fair comparison with respect to mask proposals has been resolved. The authors correctly acknowledge the need to unify mask proposals across methods and clarify that results using SAM are provided in Appendix C.2. The updated tables now include SAM-based results, which improves transparency and addresses an important fairness concern.

However, after carefully examining the revised version and the authors’ responses, AC remains unconvinced that several major concerns raised by the reviewers are fully resolved.

The design choices (negation, clustering threshold, selection logic) are still insufficiently justified. While additional explanations are provided, they do not fully address concerns about whether the proposed components are principled or primarily heuristic.

Concerns about architectural generality also remain. Reviewers explicitly requested validation beyond ViT-based CLIP models, including simple CNN-based backbones. In response, the authors only provide SigLIP experiments, which still rely on a ViT architecture. While consistent performance gains over the SigLIP baseline are observed, the absolute performance is substantially lower, and the absence of any CNN-based evaluation leaves the generality of the approach insufficiently supported.

The ablation study further raises questions about the true source of the performance gains. From Table 3(b), it appears that the majority of improvement comes from the negation and clustering steps, while the proposed global–local formulation and context token design contribute marginally.

In addition, the “opposite visualization” phenomenon is not newly identified in this paper; it was previously analyzed in CLIP Surgery. If the use of negation is a key contributor to performance, a direct comparison with CLIP Surgery-style feature or attention manipulation would be necessary to establish whether the proposed approach offers a genuinely new or superior solution rather than a simplified heuristic.

Based on the above considerations, the AC recommends rejection at this time.

**Reviewer Concerns:**

addressed: veLK
unresolved: MWds, ehKc

**Reviewer Scores:**

veLK: 2->4

---

### Decision · Program_Chairs · 2026-01-26

Reject